# Methanogenesis in the presence of oxygenic photosynthetic bacteria may contribute to global methane cycle

Jie Ye [1], Minghan Zhuang[1], Mingqiu Hong[1], Dong Zhang [1], Guoping Ren[1], Andong Hu[1], Chaohui Yang[1], Zhen He [2] ✉ & Shungui Zhou [1] ✉

Accumulating evidences are challenging the paradigm that methane in surface water primarily stems from the anaerobic transformation of organic matters. Yet, the contribution of oxygenic photosynthetic bacteria, a dominant species in surface water, to methane production remains unclear. Here we show methanogenesis triggered by the interaction between oxygenic photosynthetic bacteria and anaerobic methanogenic archaea. By introducing cyanobacterium *Synechocystis* PCC6803 and methanogenic archaea *Methanosarcina barkeri* with the redox cycling of iron, $CH_4$ production was induced in coculture biofilms through both syntrophic methanogenesis (under anoxic conditions in darkness) and abiotic methanogenesis (under oxic conditions in illumination) during the periodic dark-light cycles. We have further demonstrated $CH_4$ production by other model oxygenic photosynthetic bacteria from various phyla, in conjunction with different anaerobic methanogenic archaea exhibiting diverse energy conservation modes, as well as various common Fe-species. These findings have revealed an unexpected link between oxygenic photosynthesis and methanogenesis and would advance our understanding of photosynthetic bacteria's ecological role in the global $CH_4$ cycle. Such light-driven methanogenesis may be widely present in nature.

Atmospheric methane ($CH_4$), one of the most important greenhouse gases, reached an exceptionally high concentration of 1912 part per billion in 2022[1,2]. This calls for an immediate action to understand and address $CH_4$ emission problems. Freshwater ecosystems such as rivers, streams, lakes, oceans, and wetlands, play a vital role in contributing to the global atmospheric $CH_4$ budget[3]. It is widely recognized that $CH_4$ in freshwater ecosystems is primarily produced via the transformation of organic matters in anoxic profundal and littoral sediments[4,5]. Nevertheless, despite the limited exchange between the oxic surface layers of freshwater ecosystems and sediments due to the deep water columns, a prevalent $CH_4$ supersaturation was observed[6]. This

unexpected phenomenon, also known as the methane paradox wherein methane concentrations exceed atmospheric equilibrium values, suggests the existence of a significant $CH_4$ production process that has yet to be defined.

Photosynthetic bacteria hold a dominant presence in the surface layers of freshwater ecosystems and exhibit excellent phototactic motility and versatile metabolic patterns[7]. Their interaction with other coexisting microorganisms significantly influences the biogeochemical cycle of elements via harnessing solar light as an energy source. The correlation between photosynthetic bacteria and $CH_4$ production under illumination has been reported previously[8,9], but the underlying

[1]Fujian Provincial Key Laboratory of Soil Environmental Health and Regulation, College of Resources and Environment, Fujian Agriculture and Forestry University, Fuzhou 350002, China. [2]Department of Energy, Environmental and Chemical Engineering, Washington University in St. Louis, St. Louis, MO 63130, USA. ✉e-mail: zhenhe@wustl.edu; sgzhou@fafu.edu.cn

mechanisms are yet to be elucidated. It is likely that anoxygenic photosynthetic bacteria act as photosensitizers, driving the $CO_2$-to-$CH_4$ conversion with anaerobic methanogenic archaea when being cocultured in an anoxic layer[10]. The role of oxygenic photosynthetic bacteria in the context of $CH_4$ supersaturation is largely unknown. This oversight arises from the traditional belief that methanogenic archaea are highly sensitive to oxygen exposure and can only thrive in highly reduced, anoxic environments[11]. However, the coexistence of oxygenic photosynthetic bacteria and anaerobic methanogenic archaea occurs in various natural habitats, such as microbial mats, soil crusts, and aerobic epilimnion of an oligotrophic lake[12–14]. The in situ detection of the close attachment between methanogenic archaea and photosynthetic bacteria in these oxygenated and methane-rich environment, along with the finding that methanogens can survive oxygen exposure[15], suggested their potential interactions through direct nutrient exchange or signal transduction[16,17]. Thus, a comprehensive understanding of photosynthetically regulated $CH_4$ production is of ecological and biogeochemical importance, and will offer valuable insights into global $CH_4$ cycle with implications for climate change.

Here, we demonstrated the methanogenesis involved in the coculture of Cyanobacterium *Synechocystis* sp. strain PCC6803 (hereafter PCC6803) and *Methanosarcina barkeri* (hereafter *M. b*). PCC6803 is a model oxygenic photosynthetic bacterium that can perform solar energy conversion of water and $CO_2$ to carbohydrates and oxygen. In the absence of light, the produced carbohydrates are metabolized to generate $CO_2$ and ATP through a respiratory system, creating an anoxic microenvironment suitable for microbial methanogenesis[18]. Meanwhile, *M. b* as a model methanogen was chosen owing to its widespread environmental presence with physiological and metabolic diversity[19]. It is reported that iron exists in many open water systems and is quantitatively the most important trace metal in photosynthetic bacteria[20]. Over 99% of the dissolved Fe pool is complexed by organic ligands[21]. Therefore, Fe-ethylenediaminetetraacetic acid (Fe-EDTA) was selected as a typical iron species in this study due to its stability and solubility in aqueous solutions. The results showed that $CH_4$ production by the interaction of oxygenic photosynthetic bacteria and anaerobic methanogenic archaea was significantly enhanced through the redox cycling of Fe-ethylenediaminetetraacetic acid (Fe-EDTA), involving both syntrophic methanogenesis and abiotic methanogenesis during the periodic dark-light cycles (Fig. 1). Specifically, in darkness, the organics and $H_2$ produced by PCC6803 during dark fermentation were utilized as carbon sources and reducing equivalents by *M. b* for syntrophic methanogenesis under anoxic conditions. The significantly lowered hydrogen pressure by *M. b*, in turn, created more thermodynamically favorable conditions for PCC6803. In contrast, in illumination, the photosynthesized organic compounds and intermediate products by PCC6803 served as potential methyl donors (·$CH_3$). Meanwhile, the simultaneously produced $O_2$ stimulated reactive oxygen species (ROS) production by *M. b*. Along with the Fenton reaction with Fe-EDTA, various methyl donors were oxidized by ROS to form methyl radicals (•$CH_3$) as intermediates that eventually resulted in abiotic methanogenesis under oxic conditions. Further studies indicated that other model oxygenic photosynthetic bacteria and anaerobic methanogenic archaea were also able to conduct this light-driven methanogenesis process. These findings not only unveil an unexpected link between oxygenic photosynthesis and methanogenesis, but also advance our understanding of the ecological role of photosynthetic bacteria in the global $CH_4$ cycle.

## Results

### Light-driven methanogenesis with PCC6803 and *M. b*
PCC6803 and *M. b* were cocultured in a defined medium (Supplementary Table 1), where $CO_2$ served as the only electron acceptor. The visible light LEDs ($12 \pm 0.6 \, W \, m^{-2}$) over the wavelength range of 380–800 nm was used as a simulated sunlight source (Supplementary

Fig. 1). The experiment was conducted under a light-dark cycle of 4 hours of light and 20 hours of darkness at 35 °C but some tests were performed with a light-dark cycle of 12 h-12 h that simulates a full day. Compared to the pure *M. b* control that had a negligible $CH_4$ production under the same condition, the coculture of PCC6803 and *M. b* achieved a higher $CH_4$ yield of $1.0 \pm 0.1 \, \mu mol$. The addition of Fe-EDTA (hereafter PCC6803-*M. b*-Fe-EDTA) further enhanced the yield to $2.5 \pm 0.5 \, \mu mol$ (Fig. 2a), with a linear correlation between the $CH_4$ yield and the Fe-EDTA concentration (Supplementary Fig. 2). The single-factor experiments showed that the removal of any following components: PCC6803, *M. b*, Fe-EDTA or light, would result in a significant decline of methanogenesis performance, demonstrating the importance of each component in this process (Fig. 2a, Supplementary Fig. 3). Notably, the $CH_4$ yield with PCC6803-*M. b*-Fe-EDTA increased both under illumination and in darkness (Fig. 2b), different from the light-dependent $CH_4$ production with anoxygenic photosynthetic bacteria and methanogenic archaea that was only activated by light but inhibited in darkness[10]. This difference might be attributed to the production of $O_2$ by PCC6803 with $H_2O$ as electron donors under illumination, thereby stimulating the generation of ROS by *M. b*. These ROS, in turn, could oxidize organic matters to create potential carbon sources for biotic methanogenesis with *M. b* in the dark (see detailed discussion below). The $CH_4$ production rate with PCC6803-*M. b*-Fe-EDTA continuously increased during the three successive cycles (Fig. 2c), contributing to the rapid growth and formation of a stable syntrophic coculture for methanogenesis (Supplementary Fig. 4).

To understand the source of $CH_4$ produced by PCC6803-*M. b*-Fe-EDTA, we performed isotopic labeling experiments. The selected $m/z = 45$ ($^{13}CO_2$) and 17 ($^{13}CH_4$) were detected when $^{13}C$-labeled $NaHCO_3$ was used as the sole carbon source (Fig. 2d). This provided direct evidence that the produced $CH_4$ was derived from the $CO_2$ conversion. Unexpectedly, the relatively weak signals at $m/z = 44$ ($^{12}CO_2$) and 16 ($^{12}CH_4$) were also detected, indicating the existence of other $CH_4$ production pathways that might utilize the carbon sources synthesized during the initial cultivation process, such as biomass and oxidation intermediates (see detailed discussion below). A noticeable periodic variation in the dissolved oxygen (DO) concentration was observed in the coculture during the light-dark cycles (Fig. 2e), likely attributed to the alternate photosynthetic oxygen evolution and respiratory oxygen consumption[22]. The addition of Fe-EDTA would reduce the oxygen concentration under illumination and expedite the oxygen consumption in darkness, through the oxygen oxidation with Fe(II) chelated by EDTA, resulting in an anaerobic microenvironment for the growth and metabolism of *M. b*. This hypothesis was confirmed by the increased copy number of *mcrA* gene, which is ubiquitous in *M. b* during the methanogenesis with PCC6803-*M. b*-Fe-EDTA (Fig. 2f), along with a simultaneous rise in the copy number of *cpcG* that referred to as rod-core linker genes in PCC 6803. However, the growth of PCC6803 led to an enhanced photosynthetic oxygen evolution, elevating DO accumulation in the light and extending the time required to establish an anaerobic microenvironment for biotic methanogenesis in darkness (Fig. 2e). As a result, the rate of $CH_4$ production with PCC6803-*M. b*-Fe-EDTA slowed down somewhat after day 2 but persisted (Fig. 2b).

### Syntrophic methanogenesis by organic degradation and $CO_2$ reduction
The close contact between different microorganisms can enhance the interspecies exchange of matter and energy to shape specific communities[16]. As shown in Fig. 3a, PCC6803 and *M. b* were connected to form dense PCC6803-*M. b* biofilm at the bottom of the culture bottles after 10 days of coculturing. Both optical microscopy images (Fig. 3b) and fluorescence in situ hybridization (FISH) images (Fig. 3c) showed the aggregation of PCC6803 and *M. b* cells in the biofilm. An increase in the biofilm thickness from ~30 μm on day 0 to ~45 μm on

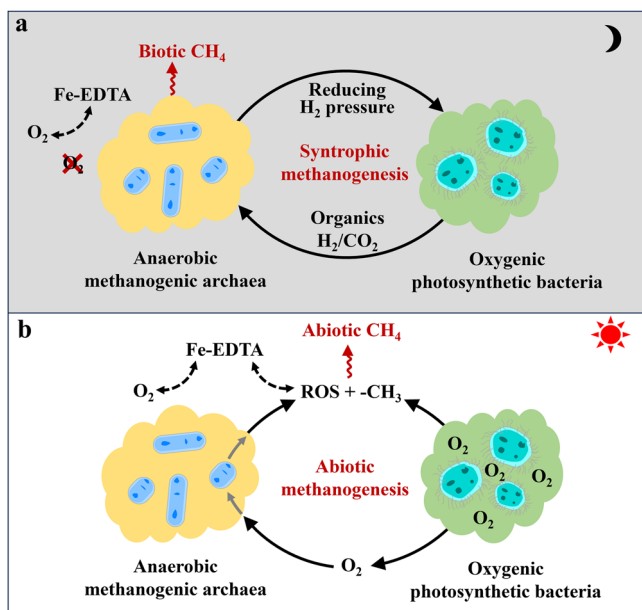

**Fig. 1 | Schematic illustration of CH₄ production in the presence of oxygenic photosynthetic bacteria. a** Syntrophic methanogenesis under anoxic conditions in darkness. **b** Abiotic methanogenesis under oxic conditions in illumination.

day 10 was observed by confocal laser scanning microscopy (CLSM) (Supplementary Fig. 5), demonstrating the growth of both microbes in the coculture biofilms.

PCC6803 can synthesize carbohydrates via photosynthesis, and then degrade these carbohydrates through a respiration process in darkness, excreting a variety of organic matters[23]. These organic matters, along with their intermediates produced via the oxidative degradation by the concomitant reactive oxygen species (ROS), would serve as the potential carbon sources for biotic methanogenesis with *M. b* (Fig. 3d). Lactate, pyruvate and acetate were detected by the [1]H nuclear magnetic resonance (NMR) spectra (Fig. 3e). These findings were further confirmed by two-dimensional NMR spectra, which provide the crucial through-bond correlations existing between the coupled protons [two-dimensional gradient-selected homonuclear correlation spectroscopy (gCOSY) in Fig. 3e] and between protons and carbons via multiple-bond correlations [gradient-selected heteronuclear multiple bond correlation (gHMBC) in Fig. 3f] for each compound[24]. The [1]H and [13]C chemical shifts of lactate, pyruvate, methanol, and acetate correspond well to values from known library spectra (PubChem Database), such as the chemical shifts of [1]H (1.96 ppm) and [13]C (176.5 ppm) in acetate, confirming the presence of these compounds in aqueous solution during syntrophic methanogenesis. Although pyruvate was reported to be the sole carbon and energy source for the growth of *M. b*[25], this species is more efficient in converting acetate to CH₄ ($CH_3COOH \rightarrow CH_4 + CO_2$) due to less energy

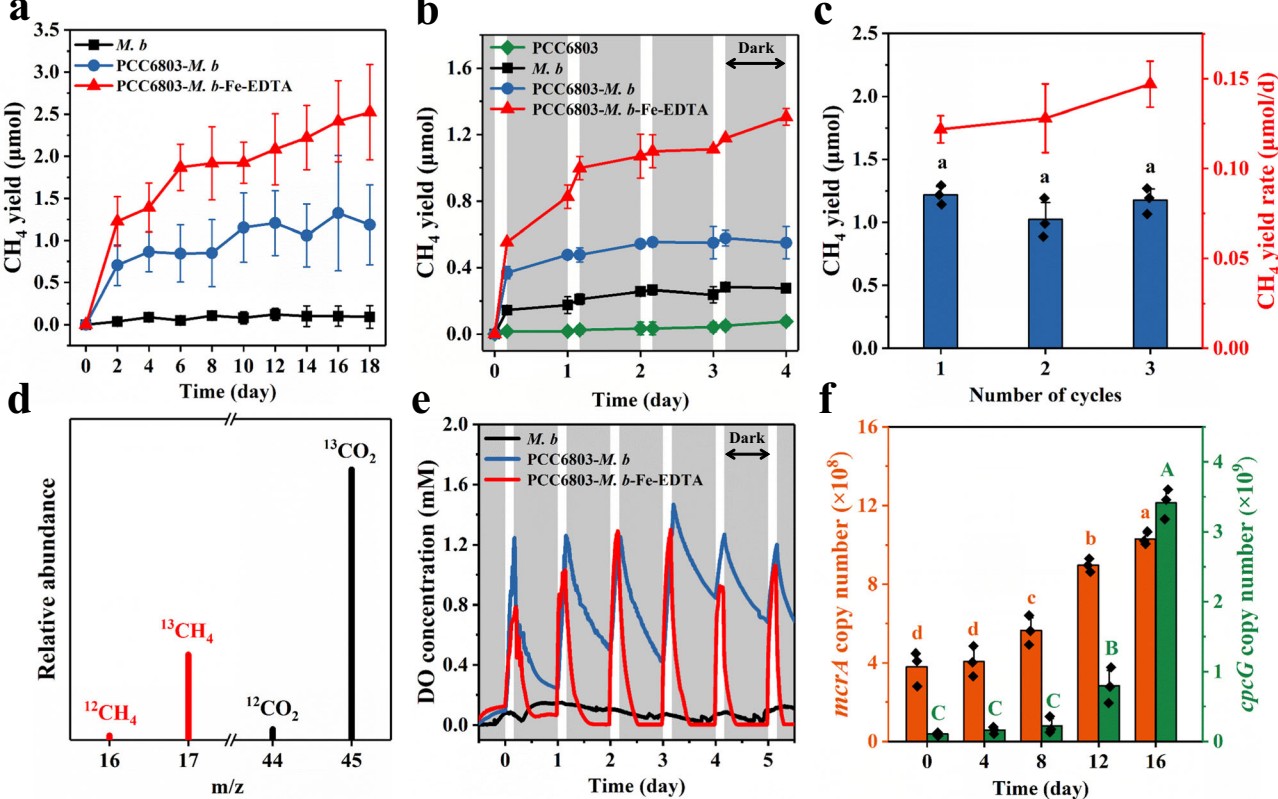

**Fig. 2 | Light-driven methanogenesis in a light-dark cycle. a** CH₄ yields by PCC6803-*M. b*-Fe-EDTA and controls. **b** Typical time course of CH₄ yield in the first 4 days. **c** CH₄ yields and yield rates during the three successive 18-day cycles. **d** Mass spectrometry of headspace gases with ¹³C-labeled NaHCO₃ as a sole carbon source. **e** Periodic variation of dissolved oxygen (DO) concentration. **f** Evaluation of the gene copy numbers of *mcrA* and *cpcG*. The gray color in **b** and **e** presents the

dark period during the light-driven methanogenesis under a light-dark cycle of 4 h-20 h. Data are presented as mean values ± SD derived from n = 3 independent experiments. Statistical analysis was conducted with paired two-tailed *t* tests, and different letters represent statistically significant difference (*P* < 0.05) in different groups. All *P* values are provided in the source data. Source data are provided as a Source Data file.

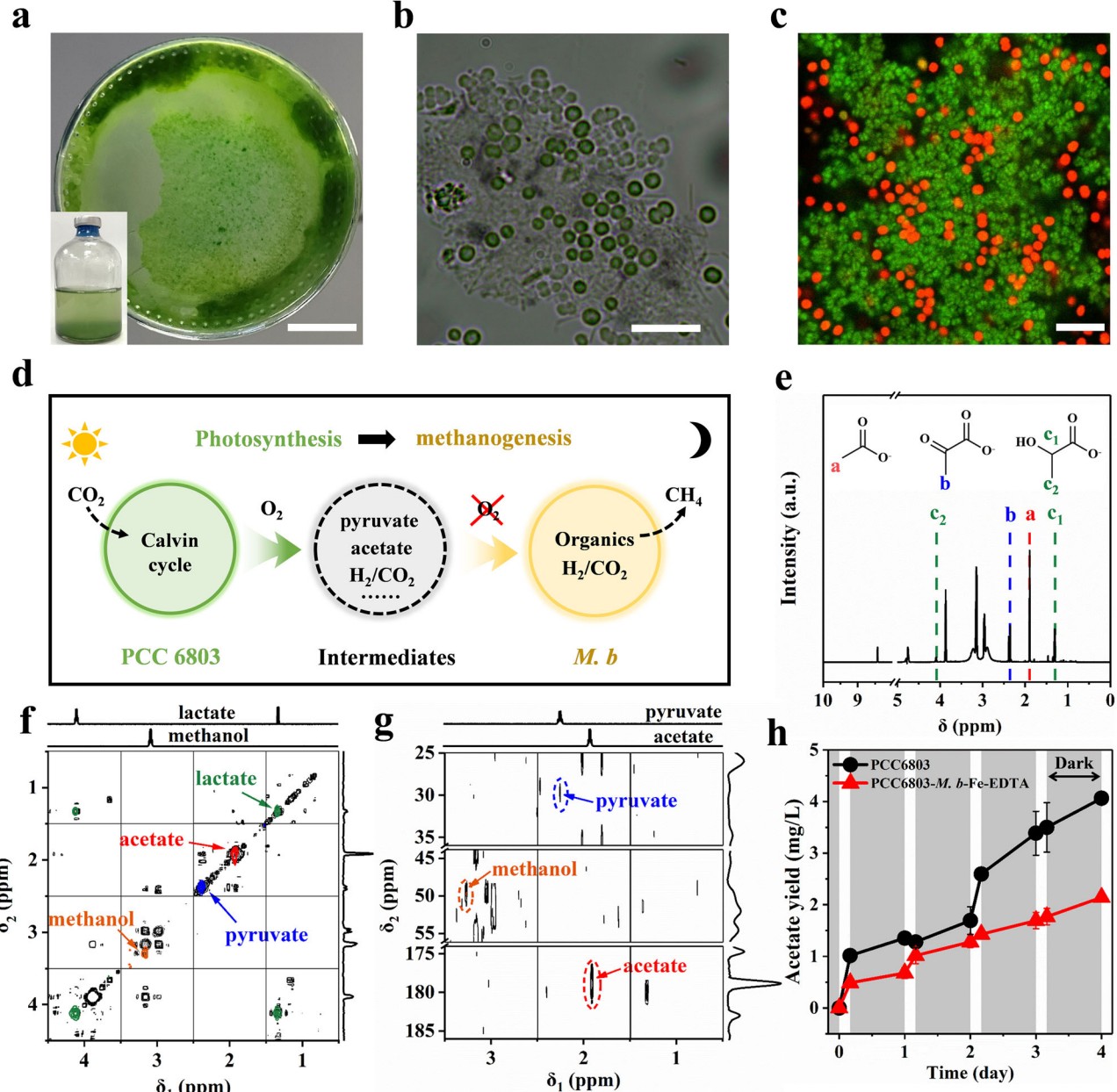

**Fig. 3 | Characterization of coculture biofilm and produced organic substances.**
**a** Dense biofilm in culture bottle (the inset image shows the picture of culture bottle). Optical microscopy image (**b**) and FISH image (**c**) with *M. b* (green-fluorescing probe) and PCC6803 (red, autofluorescence); representative of 10 images.
**d** Schematic illustration for biotic methanogenesis driven by intermediates with PCC6803-*M. b*-Fe-EDTA. **e** Possible organic substances generated via ROS oxidation. gCOSY (**f**) and gHMBC (**g**) superimposed aqueous 600-MHz NMR spectra of supernatants after the syntrophic coculture for 6 days (black), 0.1 M lactate (green), 0.1 M acetate (red), 0.1 M pyruvate (blue), and 0.1 M methanol (orange yellow). The red circled peak in (**g**) is assigned to acetate, the blue circled peak is assigned to pyruvate, and the orange yellow circled peak is assigned to methanol. **h** Variation of acetate concentration under a light-dark cycle of 4 h–20 h. The gray color presents the dark period. Data are presented as mean values ± SD derived from $n = 3$ independent experiments. Scale bars: 1 μm in (**a**), 10 μm in (**b**), and 10 μm in (**c**). Source data are provided as a Source Data file.

requirement for acetate activation[26], resulting in a lower acetate yield with PCC6803-*M. b*-Fe-EDTA than that with pure PCC6803 (Fig. 3h).

In addition to organic substances, $H_2$ was also detected in the headspace with PCC6803-*M. b*-Fe-EDTA (Supplementary Fig. 6). These $H_2$ molecules could originate from the direct (proton reduction with hydrogenase) and indirect (hydrogen release from the carbohydrate degradation) biophotolysis, and serve as electron donors for $CH_4$ production via a hydrogenotrophic pathway ($CO_2 + 4H_2 \rightarrow CH_4 + 2H_2O$)[27]. Compared with pure PCC6803, a lower $H_2$ yield was detected with PCC6803-*M. b*-Fe-EDTA after 18 days of coculture (Supplementary Fig. 6), which is probably because that $H_2$

produced by PCC6803 was used as electron donors for methanogenesis. To confirm this argument, sodium 2-bromoethanesulfonate (SBES) was added to the coculture medium to inhibit hydrogenases in methanogenic archaea. More $H_2$ residue and a lower $CH_4$ yield indicated that the hydrogenotrophic pathway was existed but inhibited by SBES in PCC6803-*M. b*-Fe-EDTA (Supplementary Fig. 7). However, it should be noted that PCC6803 contains a single [NiFe]-hydrogenase, HoxEFUYH, which is involved in fermentative hydrogen production as well as working as an electron valve when photosynthesis resumes under anaerobic conditions[28]. Notably, HoxEFUYH works bidirectionally with a bias to proton reduction rather than hydrogen

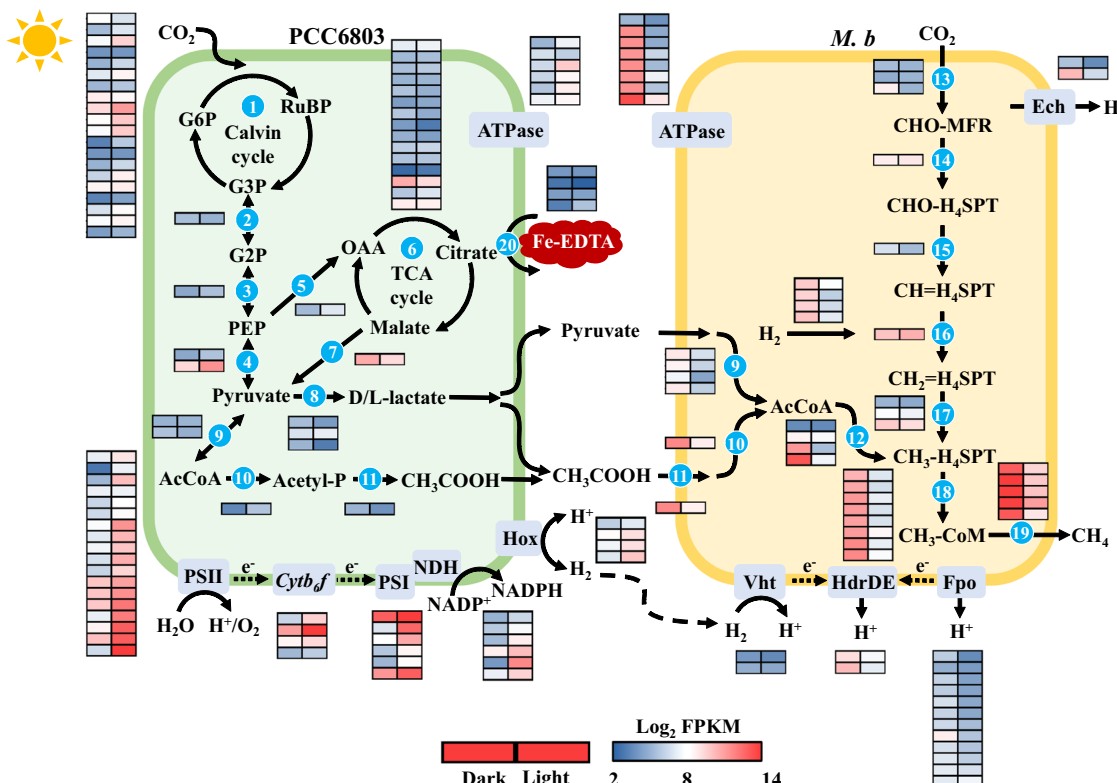

**Fig. 4 | Mechanisms of biotic methanogenesis with PCC6803-*M. b*-Fe-EDTA as revealed by transcriptomic analyses.** Genes involved in processes include those that encode (1) Calvin cycle; (2) 2;3-bisphosphoglycerate-dependent phosphoglycerate mutase Pgm; (3) enolase Eno; (4) Pyruvate kinase Pyk; (5) Phosphoenolpyruvate carboxylase Ppc; (6) TCA cycle; (7) malate dehydrogenase Mae; (8) D/L-lactate dehydrogenase D/L-LDH; (9) pyruvate-ferredoxin oxidoreductase Por; (10) acetate kinase Ack; (11) phosphate acetyltransferase Pta; (12) CO dehydrogenase/acetyl-CoA synthase Codh/Acs; (13) formylmethanofuran dehydrogenases Fmd; (14) formylmethanofuran-tetrahydromethanopterin N-formyltransferase Ftr; (15) methenyltetrahydrome-thanopterin cyclohydrolase Mch; (16) methylenetetrahydromethanopterin dehydrogenase Mtd; (17) methylenetetrahydromethanopterin reductase Mer; (18) methyltransferase subunit Mtr; (19) methyl-CoM reductase Mcr. (20) ferric uptake (Fut) and ferrous iron transport (Feo) systems. PSII photosystem II, *Cyt b₆f* cytochrome *b₆f* complex, PSI photosystem I, FNR ferredoxin−NADP⁺ reductase, NADH nicotinamide adenine dinucleotide phosphate, ATPase ATP synthase, DHAP dihydroxyacetone phosphate, RuBR ribulose-1,5-biphosphate, G3p glyceraldehyde 3-phosphate, Pep phosphoenolpyruvate, OAA oxaloacetate, AcCoA acetyl coenzyme A, Acetyl-P acetylphosphate, Fpo membrane-bound $F_{420}H_2$ dehydrogenase, Hdr heterodisulfide reductase, Hox bidirectional hydrogenase complex protein. Source data are provided as a Source Data file.

oxidation[29]. Therefore, an increase in hydrogen pressure due to hydrogen accumulation during dark fermentation might result in a significant decrease in hydrogenase activity, thereby influencing the growth and metabolism of PCC6803[30]. This inference was confirmed by the lower chlorophyll concentration, quantum yield of PSII primary photochemical reactions ($F_v/F_m$), and copy number of *cpcG* in bare PCC6803, along with a higher $H_2$ concentration compared with syntrophic methanogenesis (Supplementary Fig. 8). The superior activity of PCC6803 during syntrophic methanogenesis were attributed to the versatile metabolic pathways of *M. b*, which significantly lowered hydrogen pressure via $CO_2/H_2$ methanogenesis (Supplementary Fig. 7), and created more thermodynamically favorable conditions for the dark fermentation of PCC6803.

A distinct photocurrent was also observed under illumination during syntrophic methanogenesis (Supplementary Fig. 9a), indicating that PCC6803 can release electrons extracellularly. Although *M. b* served as electron acceptors capable of accepting photoelectrons from PCC6803 for $CO_2$ reduction ($CO_2 + 8e^- + 8H^+ \rightarrow CH_4 + 2H_2O$)[31,32], the syntrophic methanogenesis via a direct interspecies electron transfer (DIET) pathway was less likely to occur with PCC6803-*M. b*-Fe-EDTA under illumination. This was because that the growth and metabolism of *M. b* is highly sensitive to the oxygen exposure during the photosynthetic oxygen evolution. Besides the extracellular electron transfer for Fe(III) reduction, the light-induced electrons by

PCC6803 would form excited triplet state of chlorophyll in the photosystem II reaction center, which then interacts with molecular oxygen for singlet oxygen ($^1O_2$) production[33], eventually leading to the production of other ROS for abiotic methanogenesis (see detailed discussion below). In contrast, previous studies have shown that photosynthetic microorganisms can also generate an electrical current exclusively in darkness, using illumination as a recharge stage[34,35]. To validate this, two-chamber H-cells were constructed, with PCC6803 and *M. b* separately inoculated into each chamber and then electrically connected by an external circuit, to mitigate the influence of direct electron exchange between *M. b* and PCC6803 on the photoelectron measurement. Similar results were observed that a continuous current was recorded during its dark fermentation period, albeit with lesser intensity compared to that in the light (Supplementary Fig. 9b). This finding indicates the potential DIET between PCC6803 and *M. b* via an electric syntrophic coculture in darkness.

## Electron flow and energy metabolisms at the genetic level

Syntrophic methanogenesis with PCC6803-*M. b*-Fe-EDTA was further confirmed by transcriptomic analyses (Fig. 4). The transcript levels of PSII and cytochrome *b₆f* complex (*cytb₆f*) exhibited similar oscillation patterns that were highly upregulated during the light period. With plastocyanin and cyt*c₅₅₃* in the thylakoid lumen as alternate donors, the produced electrons would further transfer from *cytb₆f* to PSI,

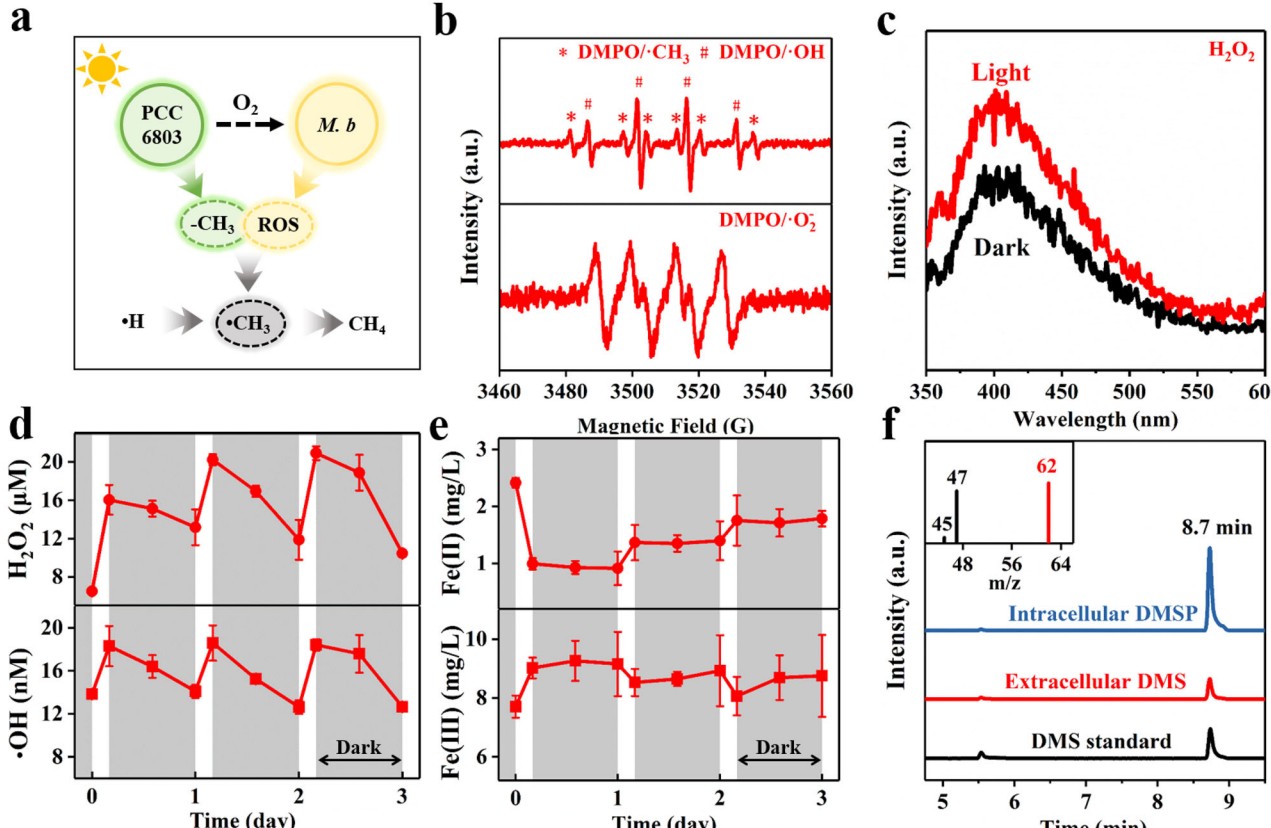

**Fig. 5 | Abiotic methanogenesis with PCC6803-*M. b*-Fe-EDTA. a** Illustration of abiotic $CH_4$ production with PCC6803-*M. b*-Fe-EDTA. **b** EPR spectra of $\cdot CH_3$, $\cdot OH$, and $\cdot O_2^-$. **c** UV/Vis absorption spectra for $H_2O_2$ with o-tolidine as the peroxide indicator. **d – e** Typical time course of the concentrations of $H_2O_2$ and $\cdot OH$ (**d**), and Fe(II) and Fe(III) (**e**). The gray color presents the dark period. **f** Characterization of DMS and DMSP as methyl donors (the inset image shows the mass spectrometry of DMS). Data are presented as mean values ± SD derived from n = 3 independent experiments. Source data are provided as a Source Data file.

resulting a higher expression of genes encoding PSI. Meanwhile, ATP synthase (ATPase) and ferredoxin-NADP⁺ reductase (FNR) were upregulated during the light period and downregulated during the dark period. The upregulation during the light period was expected, as all transcripts associated with photosynthetic electron transport and ATP synthesis were upregulated during this phase. In addition, most of the $CO_2$ fixation machinery was highly active in the light period. For example, the key genes for the Calvin cycle, such as RubisCO (*rbc*), phosphoribulokinase (*prk*) and glyceraldehyde-3-phosphate dehydrogenase (*gap*)[36], were significantly upregulated to ensure the efficient $CO_2$ fixation for producing carbohydrates (Fig. 4, Supplementary Fig. 10). The accumulated carbohydrates, acting as storage molecules, then serve as carbon sources for microbial metabolism. The oxidation of carbohydrates was further confirmed by the upregulated transcript level of genes encoding the tricarboxylic acid (TCA) cycle, such as citrate synthase (*glt*), isocitrate dehydrogenase (*icd*), synthetase (*suc*) and malate dehydrogenase (*mdh*) (Fig. 4, Supplementary Fig. 11). Likewise, genes related to the biosynthesis of lactate and acetate were significantly upregulated, which could be oxidized by ROS and used as substrates by *M. b*. Remarkably, the iron uptake and transport (ferric uptake (Fut) and ferrous iron transport (Feo) systems) were also enhanced with higher gene expression, particularly in the light period, thereby facilitating the Fe redox process during syntrophic methanogenesis.

In darkness, genes responsible for the complete pathways for oxidation of organic compounds (i.e., acetate and pyruvate) and $CO_2$ reduction were also highly expressed, suggesting the existence of multiple $CH_4$ production pathways. For instance, the significantly increased activity of pyruvate-ferredoxin oxidoreductase (Por) could

oxidize pyruvate to AcCoA and $CO_2$, and the subsequent conversion of AcCoA to $CH_4$ would occur via the native aceticlastic pathway[37]. Notably, the transcriptional levels of genes encoding functional hydrogenases for $H_2$ production and consumption (e.g., energy-converting [NiFe]-hydrogenase (Ech), methanophenazine-reducing [NiFe]-hydrogenase (Vht) and $F_{420}H_2$ dehydrogenase (Fpo))[38] were upregulated, implying that the $CO_2$ reduction was conducted via $H_2$ transfer during the dark period.

## Abiotic methanogenesis via the oxidation of methyl donors by ROS

Substantial $CH_4$ production during the light period was observed with PCC6803-*M. b*-Fe-EDTA (Fig. 2b). However, deletional control experiment revealed that PCC6803 produced almost no $CH_4$ under illumination without the presence of other factors (Supplementary Fig. 3), suggesting the potential existence of an abiotic $CH_4$ production process. ROS are vital cellular metabolic products found in all living organisms. They are responsible for oxidizing the methyl groups ($\cdot CH_3$) of methyl donors and can play a crucial role in modulating chemical $CH_4$ formation (Fig. 5a). Electron paramagnetic resonance (EPR) spectra revealed clear signals of 5,5-dimethyl-1-pyrroline-N-oxide (DMPO)/$\cdot OH$, DMPO/superoxide anion radicals ($\cdot O_2^-$) and 2,2,6,6-tetramethylpiperidine-1-oxyl (TEMPO)/$^1O_2$ from PCC6803-*M. b*-Fe-EDTA (Fig. 5b, Supplementary Fig. 12). In addition, $H_2O_2$ was produced as shown in the ultraviolet–visible (UV–Vis) absorption spectra (Fig. 5c). The concentrations of these ROS, particularly $\cdot OH$ and $H_2O_2$, significantly increased under light illumination (Fig. 5d), likely attributed to the produced $O_2$ by PCC6803 during the photosynthetic oxygen evolution that significantly stimulated the ROS production by *M. b*.

This inference was confirmed through the ROS production experiments with *M. b* under varying $O_2$ concentrations. It was found that higher $O_2$ concentrations led to the production of more ROS, such as •OH and $H_2O_2$ (Supplementary Fig. 13). Stable isotope analysis with $^{18}O_2$ further confirmed that the produced ROS stemmed from $O_2$ reduction, evidenced by the observed 5,5-dimethyl-1-pyrroline-N-oxide (DMPO)-$^{18}$OH ($m/z$ = 132.09, Supplementary Fig. 14). Previous studies have shown that when anaerobic cells were exposed to oxygen-rich environments, molecular $O_2$ adventitiously abstracted electrons from the reduced flavins or metal centers of some redox enzymes, resulting in the ROS formation ($O_2 \rightarrow •O_2^- \rightarrow H_2O_2 \rightarrow •OH$)[39–41]. As these events relied on collision frequency, the ROS production rate was directly proportional to the $O_2$ concentration[42]. Subsequently, the produced ROS were inadvertently released extracellularly[43]. Meanwhile, ROS could also be produced by PCC6803 during the photochemical energy conversion for bioenergetic production[44]. These ROS could be further used for -$CH_3$ and Fe(II) oxidization. Particularly, the light-driven $H_2O_2$ may also interact with EDTA chelated-Fe(II) for •OH production via Fenton reaction (Fe(II) + $H_2O_2 \rightarrow$ Fe(III) + •OH), leading to an increasing Fe(III) concentration under illumination in the first light-dark cycle (Fig. 5e). The produced Fe(III) through the oxidation of ROS and $O_2$ could be reduced by PCC6803 (Supplementary Fig. 15), either through intracellular metabolism or extracellular electron transfer, and finally established an effective Fe(III)/Fe(II) redox cycle.

EPR spectra also revealed the prominent signals of DMPO/•$CH_3$ (Fig. 5b). The potential sources of methyl donors for •$CH_3$ production with PCC6803-*M. b*-Fe-EDTA were diverse. On one hand, PCC6803 could synthesized organic compounds containing sulfur-bonded methyl groups, such as pyruvate and ethanol (Fig. 3g). Meanwhile, the existence of DMS was confirmed by gas chromatography-mass spectrometry (GC-MS) with an ion signal at $m/z$ = 62, along with the retention time of 8.7 min in 400 MHz $^1$H nuclear magnetic resonance (NMR) spectra (Fig. 5f). The DMS production might be partly attributed to the ROS oxidation of DMSP, because the DMSP was confirmed by the increasing intensity of DMS after alkali treatment for 12 h via the DMSP-to-DMS conversion[45]. On the other hand, the possible release of the intracellular metabolites by *M. b*, such as 2-(methylthio)ethanesulfonic acid ($CH_3$-S-CoM), might also serve as potential sources of methyl donors. These methyl donors could be oxidized by ROS for •$CH_3$ formation, ultimately leading to abiotic $CH_4$ production[46,47]. The role of ROS was further examined by the scavenger trapping tests, which revealed that the methanogenesis process was significantly inhibited after the addition of ROS quenching reagents (Supplementary Fig. 16). The $CH_4$ yield with PCC6803-*M. b*-Fe-EDTA was decreased from 2.5 μmol to 0.5 μmol with the •OH scavenger of tert-butyl alcohol (TBA, 10 mmol$^{-1}$). The results were confirmed by previous studies in which abiotic $CH_4$ production occurred with highly reactive •OH generated not only through Fenton reaction[48,49] but also through ubiquitous non-Fenton chemistry reactions driven by diversified external fields[50]. Because •OH is involved in both the organic degradation and -$CH_3$ oxidation, the difference in the $CH_4$ yield with and without the •OH scavenger suggested that the contribution of abiotic methanogenesis (i.e., methyl donors-to-$CH_4$ conversion) to total $CH_4$ production would be around 65.4%, and the remaining 34.6% of $CH_4$ came from syntrophic methanogenesis (i.e., organics/$CO_2$-to-$CH_4$ conversion). In addition, the ROS-induced •$CH_3$ could also combine with $O_2$ under illumination[47], contributing to the formation of $CH_3OH$ as a carbon source for biotic methanogenesis in darkness.

Notably, excessive ROS accumulation has been reported to induce oxidative stress, causing damage to cellular components such as DNA, proteins, and lipids, ultimately inhibiting microbial growth and survival[51]. However, ROS were effectively consumed during the light-driven methanogenesis for the oxidation of -$CH_3$ and organic compounds, as well as Fe(II) oxidation, thereby suppressing their accumulation, particularly in the dark (as shown in Fig. 5d). Additionally,

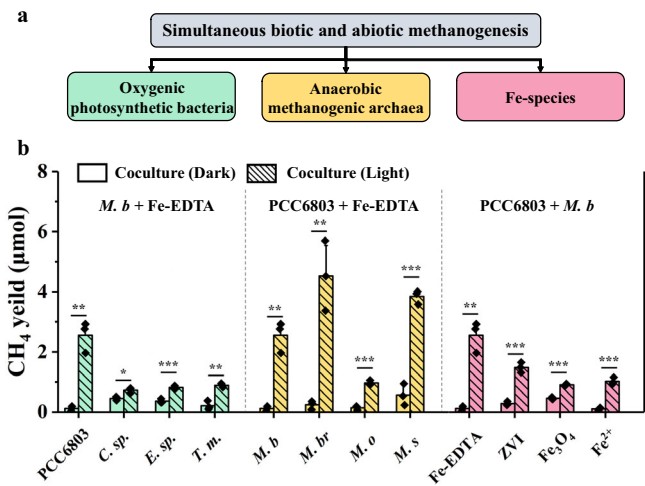

**Fig. 6 | Common light-driven methanogenesis by photosynthetic bacteria and methanogenic archaea. a** Simplified tree of influence factors on light-driven methanogenesis. **b** Light-driven methanogenesis by different oxygenic photosynthetic bacteria, anaerobic methanogenic archaea, and Fe-species (zero-valent iron (ZVI), $Fe_3O_4$ and $FeCl_2$). *C. sp.* represents *Chlorella* sp. (Chlorophyta), *E. sp.* represents *Euglena gracilis* (Euglenophyta), and *T. m* represents *Tribonema minus* (Xanthophyta). *M. b* represents *Methanosarcina barkeri*, *M. br* represents *Methanobacterium bryantii*, *M. o* represents *Methanococcoides orientis*, *M. s* represents *Methanosphaera stadtmaniae*. Data are presented as mean values ± SD derived from n = 3 independent experiments. Statistical analysis was conducted with paired two-tailed *t* tests: *$P \leq 0.05$, **$P \leq 0.01$, ***$P \leq 0.001$. All *P* values are provided in the source data. Source data are provided as a Source Data file.

the detoxification systems in both microorganisms, including superoxide dismutase, catalases, and peroxidases, were found to be significantly upregulated (Supplementary Fig. 17), effectively alleviating the potential oxidative stress and toxicity for *M. b* and PCC6803 during syntrophic methanogenesis.

## Universality of the light-driven methanogenesis in nature
The investigation of the light-driven methanogenesis was conducted to other oxygenic photosynthetic bacteria and anaerobic methanogenic archaea (Fig. 6). In addition to PCC6803 (Cyanophyta), *Tribonema minus* (Xanthophyta), *Euglena gracilis* (Euglenophyta) and *Chlorella* sp. (Chlorophyta) from various phyla also produced $CH_4$ with *M. b*. Furthermore, similar methanogenesis was observed when *M. b* was replaced with *Methanobacterium bryantii* (hydrogenotrophic methanogen), *Methanococcoides orientis* (methylotrophic methanogen), or *Methanosphaera stadtmaniae* (aceticlastic methanogen). These results suggest that the light-driven methanogenesis process with cocultures is not dependent on the energy conservation types of methanogenic archaea. We also found that various common Fe-species, such as zero-valent iron, ferric citrate and ferric oxalate, could enhance the performance of light-driven methanogenesis. Therefore, these results have provided strong evidence that the methanogenesis by oxygenic photosynthetic bacteria and anaerobic methanogenic archaea may be a prevalent occurrence in nature.

## Discussion
Unlike the previous studies that $CH_4$ production by oxygenic photosynthetic bacteria could be progressed by the demethylation of methylphosphonates or the conversion of fixed inorganic carbon into $CH_4$[52,53], this work has elucidated an unappreciated but potentially widespread pathway for $CH_4$ production. The alternating phases of photosynthetic oxygen evolution (oxic) and respiratory oxygen consumption (anoxic) are essential for methanogenesis, which could be achieved under varying light times, even a light-dark cycle of 12 h-12 h

that simulates a full day (Supplementary Fig. 18). The light-driven methanogenesis experiments were also conducted on the roof of the Research Center for Water Resources and Security Building at Fujian Agriculture and Forestry University in Fuzhou, China (latitude: 26.05 °N, longitude: 119.14 °E) under natural sunlight (from 08:00 to 20:00 with an average solar heat flux of ~0.5 kW m$^{-2}$), with ambient temperatures ranging between 25 °C and 37 °C. A similar methanogenesis process was also observed (Supplementary Fig. 19). In conclusion, besides the abiotic methanogenesis under illumination, there exists a co-evolved, specific interaction during syntrophic methanogenesis by oxygenic photosynthetic bacteria and anaerobic methanogenic archaea in darkness. Specifically, *M. b*, in the absence of other cell types except PCC6803, were benefiting from photosynthetic organic matter production. It was estimated that 5.9% of gross primary production was diverted to CH$_4$ formation. Meanwhile, due to the CO$_2$/H$_2$ methanogenesis of *M. b*, PCC6803 were benefiting from the lowered hydrogen pressure, creating more thermodynamically favorable conditions for the dark fermentation of PCC6803.

Oxygenic photosynthesis has been recognized as the most important metabolic innovation on Earth, enabling life to harness energy and reducing power directly from sunlight and water, thus liberating it from the constraints of geochemically derived reductants[54]. Consequently, these diverse and intriguing oxygenic photosynthetic bacteria contain considerable metabolic flexibility, utilizing numerous unconventional central carbon metabolic pathways and novel enzymes for autotrophic, mixotrophic, and heterotrophic growth, tailored to their specific ecological niches[22,55]. Considering the extensive coexistence and interaction of diverse microbial species in natural and engineered ecosystems[56], along with ferruginous environment on Earth (e.g., oceans with abundant Fe(II) and Fe(III)-carboxylate complexes), syntrophic methanogenesis by oxygenic photosynthetic bacteria and anaerobic methanogenic archaea creates more thermodynamically favorable conditions for both microorganisms. Thus, this light-driven methanogenesis process, involved both syntrophic methanogenesis (under anoxic conditions in darkness) and abiotic methanogenesis (under oxic conditions in illumination) during the periodic dark-light cycles, surpasses the conventional methane production pathways (i.e., acetoclastic methanogenesis and hydrogenotrophic methanogenesis), and potentially making a more significant contribution to the global CH$_4$ cycle. The inference was supported by the correlation between CH$_4$ supersaturation and photosynthesis[8,9,57]. Various potential mechanisms for CH$_4$ production by phototrophic microorganisms having been extensively investigated, including the photosynthesis-driven metabolism[9,53] and ROS-driven demethylation of methyl donors[46–49]. Our study innovatively demonstrated the synergistic interaction between these two mechanisms, along with the Fe redox cycles. However, the existence of such methanogenesis by oxygenic photosynthetic bacteria and anaerobic methanogenic archaea in the natural environments requires further validation with multiple complementary approaches, including the evaluation of in situ CH$_4$ profiles and microbial composition, incubation experiments with freshwater microbial cultures using NaH$^{13}$CO$_3$ as a supplementation carbon source, and the assessment of the exact contribution of both abiotic and biotic pathways. Meanwhile, recent studies have indicated the potential importance of various metal elements in the evolution of oxygenic photosynthesis, such as manganese[58]. Therefore, the potential involvement of other metal elements in such light-driven methanogenesis warrants further evaluation.

## Methods

### Light-driven methanogenesis experiments
PCC6803 was purchased from the China General Microbiological Culture Collection Center, and cultured in BG11 medium under illumination by visible light LEDs (12 ± 0.6 W m$^{-2}$). *M. b* MS (DSM 800) was purchased from the German Collection of Microorganisms and Cell Cultures, and cultured in heterotrophic medium modified from DSM311b medium[31,32]. When PCC6803 and *M. b* grew to their late exponential stages, cells were collected and washed with 0.9% NaCl solution for three times by centrifuging at 5000 × g for 20 min at 4 °C (Eppendorf AG 5811, Hamburg, Germany). Then, the washed cells were used to initiate coculture by inoculating PCC6803 and *M. b* cells in 200 mL of autotrophic medium with different concentrations of sterile Fe-EDTA. The suspension was sparged with sterile N$_2$/CO$_2$ (80/20, vol/vol), creating an initial anaerobic environment with CO$_2$ as the sole carbon source, and incubated without illumination for 24 h to promote heterotrophic respiration with the residual carbon sources from the initial microbial cultivation. After that, the PCC6803-*M. b*-Fe-EDTA coculture was cultivated for CH$_4$ production under a light-dark cycle of 4 h-20 h at 35 °C using visible light LEDs (12 ± 0.6 W m$^{-2}$) as light sources. For comparison, a series of single-factor experiments were conducted, by removing *M. b*, PCC6803, Fe-EDTA or light. The coculture in suspension were harvested after 18 days of light-driven methanogenesis, and the feedstocks were then replaced with the fresh sterilized autotrophic medium to start a new cycle (for a total of three cycles). To evaluate the origin of the produced CH$_4$ with PCC6803-*M. b*-Fe-EDTA, isotopic labeling experiments were conducted with $^{13}$C-labeled NaHCO$_3$ as a sole carbon source. Sodium 2-bromoethanesulfonate (SBES) as a hydrogenase inhibitor was added to evaluate the contribution of H$_2$ to the light-driven methanogenesis. ROS production experiments with *M. b* under varying O$_2$ concentrations (0.0%, 0.5% and 1.0%) were conducted to evaluate the role of *M. b* in abiotic CH$_4$ production under illumination. Meanwhile, stable isotope experiments using $^{18}$O-labeled O$_2$ were carried out to further determine the origin of the produced •OH. The DMPO-$^{18}$OH, formed through the interaction between DMPO and •$^{18}$OH, was identified using an ultrahigh-performance liquid chromatography-triple quadrupole mass spectrometer (LC–MS, TSQ Endura, Thermo Fisher, USA)[58].

To evaluate the universality of such light-driven methanogenesis in nature, several other oxygenic photosynthetic bacteria and anaerobic methanogenic archaea were selected. Oxygenic photosynthetic bacteria *Tribonema minus* (FACHB-2214, medium BG-11), *Euglena sp*. (FACHB-1862, Medium HUT) and *Chlorella sp*. (FACHB-5, medium BG-11) were obtained from the collection of Freshwater Algae Culture Collection at the Institute of Hydrobiology (FACHB), China. Anaerobic methanogenic archaea *Methanobacterium bryantii* (ATCC33272, DSM medium 1523) and *Methanosphaera stadtmaniae* (CCAM456, DSM medium 322) were purchase from Biogas Institute of Ministry of Agriculture and Rural Affairs, China. The initial inoculum of *Methanococcoides orientis* (PRJNA718391, DSM medium 141c) was graciously obtained from the laboratory of Prof. Guangyu Li in Third Institute of Oceanography, Ministry of Natural Resources, China. All culture and sampling manipulations were performed using the sterile technique.

### Microscopy and fluorescent in situ hybridization
An optical microscope (Nikon Eclipse E200, Japan) was used to examine the cocultured cells grown at their logarithmic phase. Fluorescent in situ hybridization was conducted with PCC6803-*M. b* cocultured cells that were first fixed in premixed paraformaldehyde (1%) and glutaraldehyde (0.5%), and then dehydrated in a graded ethanol series (30%, 50%, 70%, 80% and 100% for 3 min at 4 °C). Next, the samples were incubated in a UVP HL-2000 HybriLinker hybridization oven at room temperature for 2 h with PCC6803 (red, autofluorescence) and green-fluorescing probes specific for *M. b* (5′-(FAM) GTGCTCCCCCGCCAATTCCT-3′). Ultimately, the obtained samples were washed for 30 min in washing buffer, rinsed with Milli-Q water, and then visualized with a Carl Zeiss LSM 880 confocal laser scanning microscope. In addition, to investigate the variation of biofilm thickness with PCC6803-*M. b*-Fe-EDTA during light-driven methanogenesis, a graphite plate was positioned on the bottom of the culture bottle.

Then the formed biofilm was stained by a LIVE/DEAD BacLight Bacterial Viability Kit (Invitrogen, CA), and examined with a Carl Zeiss LSM 880 confocal laser scanning microscope.

## Quantitative analysis of *mcrA* and *cpcG* genes

RT-PCR was performed for the quantitative analysis of *mcrA* gene in *M. b* and *cpcG* gene in PCC6803 via a Roche LightCycler 480 System (Roche Applied Science, Penzberg, Germany). *mcrA* gene was amplified using the primer pair mcrAFor (5′-GGYGGTGTMGGDTTCACM-CARTA-3′) and mcrARev (5′- CGTTCATBGCGTAGTTVGGRTAGT-3′). *cpcG* gene was amplified using the primer pair cpcGFor (5′-GTCGGGAAGCGGGTGA-3′) and cpcGRev (5′-TTGGCGGCAGGGTTGA-3′). The detailed procedures for RNA collection and quantitative RT-PCR quantification could be found in the Supporting Information[59].

## Transcriptomic analysis

PCC6803-*M. b* cocultured cells from triplicate experiments were collected by centrifugation at $12,000 \times g$ for 2 min at 4 °C (Eppendorf AG 5811, Hamburg, Germany). Total RNA was extracted RNAprep pure cell/bacteria Kit (TIANGEN, Beijing, China), and the RNA integrity was assessed using the Agilent RNA Nano 6000 Assay Kit of the Bioanalyzer 2100 system. Library preparation for strand-specific transcriptome sequencing was generated using NEBNext Ultra II Directional RNA Library Prep Kit following the manufacturer's recommendations. All the raw sequencing data were quality-checked, and the clean reads were obtained by removing sequencing adapters, trimmed ambiguous bases (N) from the start and end, and other low-quality reads. The remaining reads were subsequently used to map against the published genome of PCC6803 (GCF_018845095.1) and *M. b* (GCF_000970025.1).

## Analytical techniques

The concentrations of $CH_4$ and $H_2$ were determined using a Shimadzu Gas Chromatograph (GC-2014, Shimadzu, Japan) equipped with both flame ionization detector (FID) and thermal conductivity detector (TCD), as well as a Porapak Q column (3.00 mm ID, 5.0 m long). Nitrogen gas (purity >99.995%) with a flow rate of 30 mL min⁻¹ was used as the carrier gas. The injector port and detector temperatures were set at 100 °C and 250 °C, respectively. The injection volume was 100 μL, with the detection limits of 0.1 ppm for $CH_4$ and 5 ppm for $H_2$. The gas products during the isotopic labeling experiments were determined by a Shimadzu GC-2010 equipped with a Shimadzu AOC-20i auto sampler system, and interfaced with a Shimadzu QP 2010S mass spectrometer (Shimadzu, Japan) (DB-5 capillary column (30 m×0.25 mm×0.25μm), Helium (99.999%) as carrier gas with a flow rate of 1.2 mL min⁻¹). The DO concentration was measured using UNISENSE OX-NP oxygen needle sensor with a detection limit of 0.3 μM, which was calibrated according to the manufacturer's instructions.

The photocurrents (*I-t*) were initially measured in a single-chamber microbial fuel cell (MFC) by applying an external potential bias of -0.5 V. Carbon cloth with a size of 3 cm × 3 cm was used as the working electrodes, and autotrophic medium with EDTA-Fe was employed as electrolyte. The cell had a liquid volume of 200 mL and a headspace volume of 150 mL. Data were recorded every 1 min by a data acquisition system (model 2700, Keithley Instruments, Ohio, USA). In addition, photocurrent measurement was also conducted under identical condition, but using two-chamber H-cells instead of the single-chamber MFC[10,60]. A proton exchange membrane (Nafion 117, DuPont Co., USA) was used to separate the anodic and cathodic chambers, along with autotrophic medium as electrolyte. To examine the compositions of organic substances produced with PCC6803-*M. b*, the coculture cells were centrifuged at $15,000 \times g$ for 20 min at 4 °C (Eppendorf AG 5811, Hamburg, Germany), and then the supernatant was characterized by the Varian INOVA 600-MHz NMR spectroscopy via ¹H NMR, gCOSY and gHMBC. The concentration of acetate was

monitored by a Shimadzu Nexis GC-2030 equipped with an FID detector and a 10 m × 0.53 mm HP-FFAP fused-silica capillary column. Nitrogen was used as the carrier gas with a flow rate of 1.0 mL min⁻¹. The reactive species were characterized using a Bruker A300-10/12 EPR spectrometer, in which •$CH_3$ and •OH were captured with DMPO while •$O_2^-$ was captured with TEMPO. The concentrations of •OH and $H_2O_2$ were measured by a terephthalic acid method and the Shimadzu UV-2600 UV–Vis spectroscopy by adding o-tolidine as the peroxide indicator, respectively. Aqueous Fe(II) concentration was determined using the ferrozine method at a wavelength of 562 nm with the Shimadzu UV-2600 UV-Vis spectroscopy[46]. Total iron ($Fe_{total}$) was measured after the reduction of Fe(III) to Fe(II) with hydroxylamine-HCl. The Fe(III) concentration was subsequently calculated as the difference between the $Fe_{total}$ and Fe(II) concentrations[61]. DMSP measurement was made by cleaving DMSP into DMS with strong alkali and quantifying DMS by gas chromatography-mass spectrometry (GCMS-QP2020 NX, Shimadzu, Japan)[62].

Cells were harvested by centrifugation at $5000 \times g$ for 10 min (Eppendorf AG 5404, Hamburg, Germany). The pellet was resuspended in 1 mL of 90 % (v/v) acetone and remained in the dark and at 4 °C for 1 h or until no green pigment was visible. Subsequently, the samples were centrifuged again at $5000 \times g$ for 10 min to separate the water soluble phase and the cell fragments (pellet) from the acetone extract (supernatant).

The quantification of Chl a was conducted with a Varian Cary Eclipse (now Agilent, USA) spectrofluorometer with the following equation[63]:

$$\text{Chl a} \ (\mu\text{g mL}^{-1}) = (11.93 \times A_{664}) - (1.93 \times A_{647}) \tag{1}$$

Where $A$ is the absorbance at different wavelengths (nm).

Cells were collected in a 1.5 mL tube and kept in the dark for 20 min. Then the photosynthetic efficiency was measured with an AquaPen-C AP-C 100 (Photon Systems Instruments, Brno, Czech Republic). The maximum quantum yield at PSII was measured in dark-adapted cells by providing a blue light (455 nm) to excite chlorophyll.

$F_v/F_m$ was calculated according to the following equation[63]:

$$F_v/F_m = (F_m - F_0)/F_m \tag{2}$$

Where $F_m$ is the maximal fluorescence yield observed in dark-adapted cells after stimulation with a saturating light pulse (3000 μmol photons m⁻² s⁻¹); $F_0$ is the fluorescence yield measured in dark-adapted sample when all PSII reaction centers are open and it was measured with a measuring light (0.045 μmol photons m⁻² s⁻¹).

The conversion efficiency of photosynthate to methane (i.e., percentage of gross primary production diverted to $CH_4$ formation) was calculated based on the reduced concentration of inorganic carbon and $CH_4$ yield.

## Reporting summary

Further information on research design is available in the Nature Portfolio Reporting Summary linked to this article.

# Data availability

The data supporting the findings of this study are available within the paper and its supplementary information. The RNA-seq data generated in this study have been deposited in the NCBI Trace Archive database under accession code PRJNA1114667. Source data are provided with this paper.

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

## Acknowledgements
This work was supported by the National Science Fund for Distinguished Young Scholars (41925028), the National Science Fund for Excellent Young Scholars (42322706), the National Natural Science Foundation of China (92251301), and a faculty startup fund from Washington University in St. Louis.

## Author contributions
J.Y., Z.H. and S.Z. conceived the idea and designed the experiments. M.Z., M.H., G.R., A.H. and C.Y. performed the experiments. J.Y., M.Z., and M.H. analyzed the data. J.Y., D.Z., Z.H. and S.Z. wrote and revised the manuscript. All the authors contributed to the interpretation of the data and preparation of the manuscript.

## Competing interests
The authors declare no competing interests.
