## [Peer Review File · Nature Communications]

Methanogenesis in the presence of oxygenic photosynthetic bacteria may contribute to global methane cycleREVIEWER COMMENTS

Reviewer #1 (Remarks to the Author):

I am honored to review this paper, which is a research direction that I am interested in and currently engaged in. In this paper, Ye and colleagues conducted an extensive study that uncovered previously overlooked connections between oxidative photosynthesis and methanogenesis. This has a beneficial impact on advancing our comprehension of the microbial and abiotic processes involved in the natural methane cycles. This research has a meticulously crafted methodology, a substantial amount of data, and a uniform writing style. Frankly speaking, I am cautiously inclined to publish this study in Nature Communications. Nevertheless, considering the current discussion, the author must clarify and revise the subsequent key issues.

Line 16-18. In the abstract, the author needs to provide a more in-depth introduction to the external biotic and abiotic factors that are responsible for the occurrence of syntrophic methanogenesis in natural environments.

Line 244-289. The generation of a significant quantity of reactive oxygen species (ROS) chemicals can potentially reach a level of toxicity for symbiotic microbe, particularly in situations when these compounds are not consumed and progressively accumulate. There is a need for additional discussions to be included here.

Line 327-348. It is advisable to take into account the metabolic diversity of the potential microorganisms and, if feasible, compare them with established methane production pathways in order to approximate the proportionate contribution of this claimed methane generation pathway in particular habitats of the natural environment.

Line 327-348. The authors should include a thorough overview of the most recent comparable accomplishments in the relevant domain, along with offering specific recommendations for future researches to validate and investigate this methane production process in specific natural environments.

Reviewer #2 (Remarks to the Author):

Syntrophic methanogenesis has received much attention in the scientific community in recent years, which provides a potentially important pathway for microbial metabolism. Particular emphasis has been placed on the electron transfer and utilization by methanogens in strictly anaerobic conditions. However, it must be admitted that the detailed information about alternative syntrophic methanogenesis by two microorganisms with different tolerance levels of oxygen for survival and reproduction has not been reported.

In this study, Zhou et al. present a novel syntrophic methanogenesis triggered by the interaction between oxygenic photosynthetic bacteria and anaerobic methanogenic archaea with the redox cycling of iron. Such syntrophic methanogenesis may be widely present in nature, and contribute to

global methane cycle, therefore with big geochemical impacts. Overall, the topic and scientific part are quite interesting and appropriate for Nature Communications. The experiments reported herein are well-designed, and the manuscript is well-written. However, the manuscript has several shortcomings that requires careful revisions. To further improve the work, I have made several comments listed below:

General Comments:

(1) Although the presented results about syntrophic methanogenesis by oxygenic photosynthetic bacteria and anaerobic methanogenic archaea are novel, some assumptions made in the manuscript are based on prior work, particularly the abiotic methanogenesis part. Therefore, some recent literatures should be cited and discussed in the manuscript to strength your conclusions, such as *J. Am. Chem. Soc.*, 2023, 145, 24590-24602; *Nat. Commun.*, 2023, 14, 4364.

(2) Overall, although there is enough detail provided for most of the methods used in this study, some ones are not sufficiently described. For instance, detection limits and method settings were not included, though these are essential for data interpretation. Please provide more details on these methods.

Specific comments:

(1) Lines 28-29, More information about the prevalent CH₄ supersaturation may be provided. under oxic conditions?

(2) Lines 44-45, If possible, more information about the coexistence of oxygenic photosynthetic bacteria and anaerobic methanogenic archaea should be provided. In addition, are there any evidences to support the statement about their potential interaction through nutrient exchange or signal transduction?

(3) Line 62, Why did you choose Fe-ethylenediaminetetraacetic acid (Fe-EDTA) as Fe species for experiments? Please rationalize these choices.

(4) Line 80, More information for the applied light source for illumination should be provided, such as the wavelength spectra for the visible light LEDs.

(5) Lines 93-94, Why the methanogenesis behaviour was different between anoxygenic photosynthetic bacteria and oxygenic photosynthetic bacteria under the light-dark conditions?

(6) Lines 97-98, Is it possible to provide more evidences to support the statement about the rapid growth and formation of a stable syntrophic coculture for methanogenesis?

(7) Lines 130-132, Can you show the change of biofilm during the coculture in a time scale with different pictures, maybe in Supporting Information? This may give the readers with more intuitive impression.

(8) Lines 144-146, Maybe more discussion about the results of two-dimensional NMR should be conducted.

(9) Lines 165-170, Is it possible for *M. b* to directly use the electrons from the respiration process of PCC6803 in darkness for methanogenesis? I can understand that it may be very difficult to provide the direct evidences, but you can add the related discussion in the content.

(10) Lines 274-275, More information and discussion should be replenished for Fig. 5f.

(11) Based on the discussion in the manuscript, it is better to adjust the order of different parameters in Figure 6 (First oxygenic photosynthetic bacteria, then anaerobic methanogenic archaea, and finally Fe-species). Meanwhile, the related experimental conditions should be more clearly clarified. For instance, which kind of Fe-species did you use when you investigated the influence of different oxygenic photosynthetic bacteria on the performance of syntrophic methanogenesis?

(12) Line 335, The related information about the natural sunlight should be replenished.

Reviewer #3 (Remarks to the Author):

This is a very interesting and very complex article. Stepping back and taking a broad perspective on the findings, the authors conclude from their experiments that a variety of methanogenic bacteria produce methane in the light and in the dark when co-cultured with a variety of different oxygenic phototrophs. More methane is made in the light (Fig. 2b), and FeEDTA considerably enhanced methane production. I suggest the authors choose among the good ideas they report here and build stronger experimental support for the ideas they want to prioritize. It would be a significant contribution to prove a general mechanism that supports a partnership between methanogens and oxygenic photoautotrophs. Presumably (but not explicitly clear in this manuscript), part of this partnership is the phototrophs benefit from the methanogens when they deplete oxygen by respiration. In addition to this concept, also in this manuscript are the concepts of direct interspecies electron transfer via Fe-EDTA, and abiotic methanogenesis via ROS. These three concepts all have some support from the reported experiments but none has compelling experimental support. In a manuscript with this many complex ideas and experiments, the supporting documentation can be very important for explaining concepts when page length limitations prevent digression. But, the supporting documents include interesting figures with very little explanation. Overall, I find the ideas in this manuscript interesting but the presentation of the data made it difficult to fully evaluate the experimental support the author's ideas. Although I think clarity and focus are a problem, I suspect with greater clarity and focus I would ask for more experiments. At this time I don't want to ask for more experiments when I'm not sure which if the findings are most central.

1. The authors refer to the relationship between phototrophs and methanogen as a syntrophy but I did not notice evidence supporting the idea that the phototrophs benefited. In fact, in Fig 2e we see that the introduction of methanosarcina (M.b.) +Fe-EDTA seems to accelerate O₂ decline (note there is no PCC6308 control shown for comparison). My interpretation is that M.b. is consuming oxygen, probably resulting in more rapid depletion of O₂ by the cyanobacteria when they are respiring in the dark, which likely will have negative consequences for the cyanobacteria. How does the cyanobacterium benefit? The impact appears to be negative. I recommend to not use the term "syntrophy" unless you can show benefits to the cyanobacterial partner. I appreciate that it is possible that the methanogens are making energy-yielding anaerobic metabolism more favorable for the cyanobacteria when O₂ is exhausted, by lowering H₂ partial pressures, but I found the discussion of that important mechanism too brief and unconvincing.

2. A rather straightforward, conventional explanation for methanogenic activity would be that the methanogens, in the absence of other cell types except the photoauxotrophs, are benefiting from photosynthetic organic matter production, performing methanogenesis when oxygen is depleted by cyanobacterial respiration in the dark. While not surprising from a theoretical perspective, this potentially is an interesting finding. What percentage of gross primary production was diverted to methane formation? Are your findings a manifestation of common microbiological activities, or is

there a co-evolved, specific interaction occurring? I found myself unable to decide which was the case for the data provided.

I recommend that you estimate the percentage of gross primary production accounted for by methane production. Is there evidence that anaerobic PCC6308 can use fermentative metabolism to produce energy in the absence of oxygen, which would indicate they might benefit from the methanogens creating thermodynamically favorable conditions for fermentation?

3. The manuscript indicates that Fe-EDTA supports methanogenesis, presumably by serving as a carrier for electrons DIET. Can you prove that PCC6803 reduces Fe-EDTA and Mb. oxidizes it?

4. Figure 4. “Mechanisms of biotic methanogenesis with PCC6803-M. b-Fe-EDTA as revealed by transcriptomic analyses”. It’s confusing to show ROS in this figure but no role for Fe-EDTA, which is purportedly important for biological methane production.

5. In Fig.2, oxygen production appears to continue for 5 days (2e) but methane production falls off sharply after day 2. Is this the correct interpretation of the data? Why does methane production stop?

6. There is almost no explanation of the term “photocurrent”, and the description of these experiments is too brief. If the photocurrent is through a graphite substrate, does this mean its competing with electron transfer to M.b.?

Legend, Fig. 2. Indicate that grey means dark, if that is correct.

The legends for the supporting figures are too brief. Please add details.

Re: NCOMMS-24-04041

Point-by-point responses to Referees' Comments

Reviewer #1 (Remarks to the Author):

I am honored to review this paper, which is a research direction that I am interested in and currently engaged in. In this paper, Ye and colleagues conducted an extensive study that uncovered previously overlooked connections between oxidative photosynthesis and metagenesis. This has a beneficial impact on advancing our comprehension of the microbial and abiotic processes involved in the natural methane cycles. This research has a meticulously crafted methodology, a substantial amount of data, and a uniform writing style. Frankly speaking, I am cautiously inclined to publish this study in Nature Communications. Nevertheless, considering the current discussion, the author must clarify and revise the subsequent key issues.

We really appreciate the referee's insightful suggestions to help us significantly improve the quality of our manuscript. We have carefully revised the manuscript, and sincerely hope that our revisions have satisfactorily addressed the referee's concerns.

1) Line 16-18. In the abstract, the author needs to provide a more in-depth introduction to the external biotic and abiotic factors that are responsible for the occurrence of syntrophic methanogenesis in natural environments.

Response: We thank the referee for pointing out this issue. As suggested, a more in-depth introduction to the external biotic and abiotic factors that are responsible for the occurrence of syntrophic methanogenesis in natural environments has been replenished in the revised manuscript to the strength the conclusion.

Revised in text: "We further demonstrated CH₄ production by other model oxygenic photosynthetic bacteria from various phyla, in conjunction with different anaerobic methanogenic archaea exhibiting diverse energy conservation modes, as well as various common Fe-species. Therefore, such syntrophic methanogenesis may be widely present in nature. These findings have revealed an unexpected link between oxygenic photosynthesis and methanogenesis, and would advance our understanding of photosynthetic bacteria's ecological role in the global CH₄ cycle."

2) Line 244-289. The generation of a significant quantity of reactive oxygen species (ROS) chemicals can potentially reach a level of toxicity for symbiotic microbe, particularly in situations when these

compounds are not consumed and progressively accumulate. There is a need for additional discussions to be included here.

Response: We thank the referee for pointing out this issue. As suggested, additional discussions about the accumulation and consumption of produced ROS on the growth of symbiotic microbe have been included in the revised manuscript.

Revised in text: “Notably, excessive ROS accumulation has been reported to induce oxidative stress, causing damage to cellular components such as DNA, proteins, and lipids, ultimately inhibiting microbial growth and survival.⁴⁸ However, ROS were effectively consumed during syntrophic methanogenesis for the oxidation of -CH₃ and organic compounds, as well as Fe (II) oxidation, thereby suppressing their accumulation, particularly in the dark (as shown in Fig. 5d). Additionally, the detoxification systems in both microorganisms, including superoxide dismutase, catalases, and peroxidases, were found to be significantly upregulated (Supplementary Fig. 15), effectively alleviating the potential oxidative stress and toxicity for *M. b* and PCC6803 during syntrophic methanogenesis.”.

Supplementary Fig. 15 Transcriptomic analyses of key antioxidant stress genes in *M. b* and PCC6803 during the syntrophic methanogenesis. Sod, superoxide dismutase; Kat, catalase; sll1621, type II peroxiredoxins; AhpC and AhpE, alkylhydroperoxidases; Htp and Hsp, heat shock proteins; Rbr, rubrerythrin; Dps, DNA-binding protein from starved cells.

Reference

Hong, Y., Zeng, J., Wang, X., Drlica, K., Zhao, X. Post-stress bacterial cell death mediated by reactive oxygen species *P. Natl. Acad. Sci.* **116**, 10064-10071 (2019).

3) Line 327-348. It is advisable to take into account the metabolic diversity of the potential

microorganisms and, if feasible, compare them with established methane production pathways in order to approximate the proportionate contribution of this claimed methane generation pathway in particular habitats of the natural environment.

Response: We appreciate the reviewer's valuable suggestion. As suggested, the relevant discussion has been replenished in the revised manuscript.

Revised in text: “Oxygenic photosynthesis has been recognized as the most important metabolic innovation on Earth, enabling life to harness energy and reducing power directly from sunlight and water, thus liberating it from the constraints of geochemically derived reductants.⁵¹ Consequently, these diverse and intriguing oxygenic photosynthetic bacteria exist considerable metabolic flexibility, utilizing numerous unconventional central carbon metabolic pathways and novel enzymes for autotrophic, mixotrophic, and heterotrophic growth, tailored to their specific ecological niches.^{22,52} Considering the extensive coexistence and interaction of diverse microbial species in natural and engineered ecosystems,⁵³ along with ferruginous environment on Earth (e.g., oceans with abundant Fe(II) and Fe(III)-carboxylate complexes), syntrophic methanogenesis by oxygenic photosynthetic bacteria and anaerobic methanogenic archaea creates more thermodynamically favourable conditions for both microorganisms. Thus, this syntrophic methanogenesis process surpasses the conventional methane production pathways (i.e., acetoclastic methanogenesis and hydrogenotrophic methanogenesis), and potentially making a more significant contribution to the global CH₄ cycle.”.

References

22. Saper, G., *et al.* Live cyanobacteria produce photocurrent and hydrogen using both the respiratory and photosynthetic systems. *Nat. Commun.* **9**, 2168 (2018).

51. Fischer, W. W., Hemp, J., Johnson, J. E. Evolution of oxygenic photosynthesis. *Annu. Rev. Earth Pl. Sc.* **44**, 647-683 (2016).

52. Tang, K.-H., Tang, Y. J., Blankenship, R. E. Carbon metabolic pathways in phototrophic bacteria and their broader evolutionary implications. *Front. Microbiol.* **2**, 11061 (2011).

53. Kato, S., Watanabe, K. Ecological and evolutionary interactions in syntrophic methanogenic consortia. *Microbes Environ.* **25**, 145-151 (2010).

4) *Line 327-348. The authors should include a thorough overview of the most recent comparable accomplishments in the relevant domain, along with offering specific recommendations for future researches to validate and investigate this methane production process in specific natural environments.*

Response: We appreciate the reviewer's valuable suggestion. As suggested, the relevant information has been replenished in the revised manuscript.

Revised in text: “The inference was supported by the correlation between CH₄ supersaturation and photosynthesis.^{8,9,54} Various potential mechanisms for CH₄ production by phototrophic microorganisms having been extensively investigated, including the photosynthesis-driven metabolism^{9,50} and ROS-driven demethylation of methyl donors.⁴³⁻⁴⁶ Our study innovatively demonstrated the synergistic interaction between these two mechanisms, along with the Fe redox cycles. However, the existence of such syntrophic methanogenesis by oxygenic photosynthetic bacteria and anaerobic methanogenic archaea in the natural environments requires further validation with multiple complementary approaches, including the evaluation of *in situ* CH₄ profiles and microbial composition, incubation experiments with freshwater microbial cultures using NaH¹³CO₃ as a supplementation carbon source, and the assessment of the exact contribution of both abiotic and biotic pathways. Meanwhile, recent studies have indicated the potential importance of various metal elements in the evolution of oxygenic photosynthesis, such as manganese.⁵⁵ Therefore, the potential involvement of other metal elements in such syntrophic methanogenesis warrants further evaluation.”.

References

8. Fixen, K. R., *et al.* Light-driven carbon dioxide reduction to methane by nitrogenase in a photosynthetic bacterium. *P. Natl. Acad. Sci.* **113**, 10163-10167 (2016).
9. Perez-Coronel, E., Michael Beman, J. Multiple sources of aerobic methane production in aquatic ecosystems include bacterial photosynthesis. *Nat. Commun.* **13**, 6454 (2022).
43. Ernst, L., *et al.* Methane formation driven by reactive oxygen species across all living organisms. *Nature* **603**, 482-487 (2022).
44. Althoff, F., *et al.* Abiotic methanogenesis from organosulphur compounds under ambient conditions. *Nat. Commun.* **5**, 4205 (2014).
45. Hädeler, J., Velmurugan, G., Lauer, R., Radhamani, R., Keppler, F., Comba, P. Natural abiotic iron-oxido-mediated formation of C1 and C2 compounds from environmentally important methyl-substituted substrates. *J. Am. Chem. Soc.* **145**, 24590-24602 (2023).
46. Ernst, L., *et al.* Methane formation driven by light and heat prior to the origin of life and beyond. *Nat. Commun.* **14**, 4364 (2023).
50. Bižić, M., *et al.* Aquatic and terrestrial cyanobacteria produce methane. *Sci. Adv.* **6**, 5343 (2020).
54. Deines, P., Hammerschmidt, K., Bosch, T. C. Microbial species coexistence depends on the host environment. *mBio* **11**, 10-1128 (2020).
55. Huang, L., Liu, X., Rensing, C., Yuan, Y., Zhou, S., Nealson, K. H. Light-independent anaerobic microbial oxidation of manganese driven by an electrosynthetic coculture. *ISME J.* **17**, 163-171 (2022).

Reviewer #2 (Remarks to the Author):

Syntrophic methanogenesis has received much attention in the scientific community in recent years, which provides a potentially important pathway for microbial metabolism. Particular emphasis has been placed the electron transfer and utilization by methanogens in strictly anaerobic conditions. However, it must be admitted that the detailed information about alternative syntrophic methanogenesis by two microorganisms with different tolerance levels of oxygen for survival and reproduction has not been reported.

In this study, Zhou et al. present a novel syntrophic methanogenesis triggered by the interaction between oxygenic photosynthetic bacteria and anaerobic methanogenic archaea with the redox cycling of iron. Such syntrophic methanogenesis may be widely present in nature, and contribute to global methane cycle, therefore with big geochemical impacts. Overall, the topic and scientific part are quite interesting and appropriate for Nature Communications. The experiments reported herein are well-designed, and the manuscript is well-written. However, the manuscript has several shortcomings that requires careful revisions. To further improve the work, I have made several comments listed below:

We really appreciate the referee's kind suggestions to help us further improve the quality of our manuscript. We have carefully revised the manuscript, and sincerely hope that our revisions have satisfactorily addressed the referee's concerns.

General Comments:

(1) Although the presented results about syntrophic methanogenesis by oxygenic photosynthetic bacteria and anaerobic methanogenic archaea are novel, some assumptions made in the manuscript are based on prior work, particularly the abiotic methanogenesis part. Therefore, some recent literatures should be cited and discussed in the manuscript to strength your conclusions, such as J. Am. Chem. Soc., 2023, 145, 24590-24602; Nat. Commun., 2023, 14, 4364.

Response: We appreciate the reviewer's valuable suggestion. As suggested, the recent literatures have been cited and discussed in the revised manuscript to the strength the conclusion.

Revised in text: "These methyl donors could be oxidized by ROS for $\bullet\text{CH}_3$ formation, ultimately leading to abiotic CH_4 production.^{43,44}".

"In addition, the ROS-induced $\bullet\text{CH}_3$ could also combine with O_2 under illumination,⁴⁴ contributing to the formation of CH_3OH as a carbon source for biotic methanogenesis in darkness.".

Revised in text: “The results were confirmed by previous studies, where abiotic CH₄ production occurred with highly reactive •OH generated not only through Fenton reaction^{45,46} but also through ubiquitous non-Fenton chemistry reactions driven by diversified external fields.⁴⁷”.

References

43. Ernst, L., *et al.* Methane formation driven by reactive oxygen species across all living organisms. *Nature* **603**, 482-487 (2022).

44. Althoff, F., *et al.* Abiotic methanogenesis from organosulphur compounds under ambient conditions. *Nat. Commun.* **5**, 4205 (2014).

45. Hädel, J., Velmurugan, G., Lauer, R., Radhamani, R., Keppler, F., Comba, P. Natural abiotic iron-oxido-mediated formation of C1 and C2 compounds from environmentally important methyl-substituted substrates. *J. Am. Chem. Soc.* **145**, 24590-24602 (2023).

46. Ernst, L., *et al.* Methane formation driven by light and heat prior to the origin of life and beyond. *Nat. Commun.* **14**, 4364 (2023).

47. Jie, Y., *et al.* Abiotic methane production driven by ubiquitous non-Fenton-type reactive oxygen species. *Angew. Chem. Int. Ed.* e202403884 (2024).

(2) Overall, although there is enough detail provided for most of the methods used in this study, some ones are not sufficiently described. For instance, detection limits and method settings were not included, though these are essential for data interpretation. Please provide more details on these methods.

Response: We thank the referee for pointing out this issue. As suggested, more detailed information has been replenished in the revised manuscript.

Revised in text: “The concentrations of H₂ and CH₄ were determined using a Shimadzu Gas Chromatograph (GC-2014, Shimadzu, Japan) equipped with both flame ionization detector (FID) and thermal conductivity detector (TCD), as well as a Porapak Q column (3.00 mm ID, 5.0 m long). Nitrogen gas (purity >99.995%) with a flow rate of 30 mL min⁻¹ was used as the carrier gas. The injector port and detector temperatures were set at 100 °C and 250 °C, respectively. The injection volume was 100 µL, with the detection limits of 0.1 ppm for CH₄ and 5 ppm for H₂.”.

“The gas products during the isotopic labelling experiments were determined by a Shimadzu GC-2010 equipped with a Shimadzu AOC-20i auto sampler system, and interfaced with a Shimadzu QP 2010S mass spectrometer (Shimadzu, Japan) (DB-5 capillary column (30 m × 0.25 mm × 0.25 µm), Helium (99.999%) as carrier gas with a flow rate of 1.2 mL min⁻¹).”.

“The DO concentration was measured using UNISEN SE OX NP oxygen needle sensor with a

detection limit of 0.3 μM , which was calibrated according to the manufacturer's instructions.”.

“The photocurrents ($I-t$) were initially measured in a single-chamber microbial fuel cell (MFC) by applying an external potential bias of -0.5 V. Carbon cloth with a size of 3 cm \times 3 cm was used as the working electrodes, and autotrophic medium with EDTA-Fe was employed as electrolyte. The cell had a liquid volume of 200 mL and a headspace volume of 150 mL. Data were recorded every 1 min by a data acquisition system (model 2700, Keithley Instruments, Ohio, USA). In addition, photocurrent measurement was also conducted under identical condition, but using two-chamber H-cells instead of the single-chamber MFC.¹⁰ A proton exchange membrane (Nafion 117, DuPont Co., USA) was used to separate the anodic and cathodic chambers, along with autotrophic medium as electrolyte.”.

“The concentration of acetate was monitored by a Shimadzu Nexis GC-2030 equipped with an FID detector and a 10 m \times 0.53 mm HP-FFAP fused-silica capillary column. Nitrogen was used as the carrier gas with a flow rate of 1.0 mL min^{-1} .”.

“Aqueous Fe(II) concentration was determined using the ferrozine method at a wavelength of 562 nm with the Shimadzu UV-2600 UV-Vis spectroscopy.⁴³ Total iron (Fe_{total}) was measured after the reduction of Fe(III) to Fe(II) with hydroxylamine-HCl. The Fe(III) concentration was subsequently calculated as the difference between the Fe_{total} and Fe(II) concentrations.⁵⁷”.

“The syntrophic methanogenesis experiments was also conducted on the roof of the Research Center for Water Resources and Security Building at Fujian Agriculture and Forestry University in Fuzhou, China (latitude: 26.05 °N, longitude: 119.14 °E) under natural sunlight (from 08:00 to 20:00 with an average solar heat flux of $\sim 0.5 \text{ kW m}^{-2}$), with ambient temperatures ranging between 25 °C and 37 °C.”.

“In addition, to investigate the variation of biofilm thickness with PCC6803-*M. b*-Fe-EDTA during syntrophic methanogenesis, a graphite plate was positioned on the bottom of the culture bottle. Then the formed biofilm was stained by a LIVE/DEAD BacLight Bacterial Viability Kit (Invitrogen, CA), and examined with a Carl Zeiss LSM 880 confocal laser scanning microscope.”.

“Cells were harvested by centrifugation at 5000 \times g for 10 min (Eppendorf AG 5404, Hamburg, Germany). The pellet was resuspended in 1 mL of 90 % (v/v) acetone and remained in the dark and at 4 °C for 1 h or until no green pigment was visible. Subsequently, the samples were centrifuged again at 5000 \times g for 10 min to separate the water soluble phase and the cell fragments (pellet) from the acetone extract (supernatant).

The quantification of Chl a was conducted with a Varian Cary Eclipse (now Agilent, USA) spectrofluorometer with the following equation:⁵⁹

$$\text{Chl a } (\mu\text{g mL}^{-1}) = (11.93 \times A_{664}) - (1.93 \times A_{647})$$

Where A is the absorbance at different wavelengths (nm).”.

“Cells were collected in a 1.5 mL tube and kept in the dark for 20 min. Then the photosynthetic efficiency was measured with an AquaPen-C AP-C 100 (Photon Systems Instruments, Brno, Czech Republic). The maximum quantum yield at PSII was measured in dark-adapted cells by providing a blue light (455 nm) to excite chlorophyll.

Fv/Fm was calculated according to the following equation:⁵⁹

$$Fv/Fm = (Fm - F0)/Fm$$

Where Fm is the maximal fluorescence yield observed in dark-adapted cells after stimulation with a saturating light pulse (3000 $\mu\text{mol photons m}^{-2} \text{ s}^{-1}$); F0 is the fluorescence yield measured in dark-adapted sample when all PSII reaction centers are open and it was measured with a measuring light (0.045 $\mu\text{mol photons m}^{-2} \text{ s}^{-1}$).

The conversion efficiency of photosynthate to methane (i.e., percentage of gross primary production diverted to CH₄ formation) was calculated based on the reduced concentration of inorganic carbon and CH₄ yield.”.

Specific comments:

(1) Lines 28-29, More information about the prevalent CH₄ supersaturation may be provided. under oxic conditions?

Response: We thank the referee for his/her valuable comment. As suggested, more information about the prevalent CH₄ supersaturation has been replenished in the revised manuscript.

Revised in text: “Nevertheless, despite the limited exchange between the oxic surface layers of freshwater ecosystems and sediments due to the deep water columns, a prevalent CH₄ supersaturation was observed.⁶ This unexpected phenomenon, also known as the methane paradox wherein methane concentrations exceed atmospheric equilibrium values, suggests the existence of a significant CH₄ production process that has yet to be defined.”.

Reference

6. Mao, Y., *et al.* Aerobic methane production by phytoplankton as an important methane source of aquatic ecosystems: Reconsidering the global methane budget. *Sci. Total Environ.* **907**, 167864 (2023).

(2) Lines 44-45, If possible, more information about the coexistence of oxygenic photosynthetic bacteria and anaerobic methanogenic archaea should be provided. In addition, are there any evidences to support the statement about their potential interaction through nutrient exchange or signal transduction?

Response: We thank the referee for pointing out this issue. more information and references have been replenished in the revised manuscript to support the relevant statements.

Revised in text: “However, the coexistence of oxygenic photosynthetic bacteria and anaerobic methanogenic archaea occurs in various natural habitats, such as microbial mats, soil crusts, and aerobic epilimnion of an oligotrophic lake.¹²⁻¹⁴ The *in-situ* detection of the close attachment between methanogenic archaea and photosynthetic bacteria in these oxygenated and methane-rich environment, along with the finding that methanogens can survive oxygen exposure,¹⁵ suggested their potential interactions through direct nutrient exchange or signal transduction.^{16,17}”.

References

15. Yuan, Y., Conrad, R., Lu, Y. Responses of methanogenic archaeal community to oxygen exposure in rice field soil. *Environ. Microbiol. Rep.* **1**, 347-354 (2009).

16. Kouzuma, A., Watanabe, K. Exploring the potential of algae/bacteria interactions. *Curr. Opin. Biotech.* **33**, 125-129 (2015)

17. Bandyopadhyay, A., Stöckel, J., Min, H., Sherman, L. A., Pakrasi, H. B. High rates of photobiological H₂ production by a cyanobacterium under aerobic conditions. *Nat. Commun.* **1**, 139 (2010).

(3) Line 62, Why did you choose Fe-ethylenediaminetetraacetic acid (Fe-EDTA) as Fe species for experiments? Please rationalize these choices.

Response: We are sorry that the statements in the original manuscript were not clear. iron exists in many open water systems and is quantitatively the most important trace metal in photosynthetic bacteria (*Photosynth. Res.* 60, 111-150 (1999)). Notably, over 99% of this dissolved Fe pool is complexed by organic ligands (*Nat. Geosci.* 3, 675-682 (2010)). Therefore, Fe-ethylenediaminetetraacetic acid (Fe-EDTA) was selected as a typical iron species in this study due to its stability and solubility in aqueous solutions, making it suitable for controlled laboratory experiments. The relevant information has been replenished in the revised manuscript.

Revised in text: “It is reported that iron exists in many open water systems and is quantitatively the most important trace metal in photosynthetic bacteria.²⁰ Over 99% of the dissolved Fe pool is complexed by organic ligands.²¹ Therefore, Fe-ethylenediaminetetraacetic acid (Fe-EDTA) was

selected as a typical iron species in this study due to its stability and solubility in aqueous solutions.”.

References

20. Raven, J. A., Evans, M. C. W., Korb, R. E. The role of trace metals in photosynthetic electron transport in O₂-evolving organisms. *Photosynth. Res.* **60**, 111-150 (1999).

21. Boyd, P. W., Ellwood, M. J. The biogeochemical cycle of iron in the ocean. *Nat. Geosci.* **3**, 675-682 (2010).

(4) Line 80, More information for the applied light source for illumination should be provided, such as the wavelength spectra for the visible light LEDs.

Response: We thank the referee for pointing out this issue. The relevant information has been replenished in the revised manuscript.

Revised in text: “The visible light LEDs ($12 \pm 0.6 \text{ W m}^{-2}$) over the wavelength range of 380–800 nm was used as a simulated sunlight source (Supplementary Fig. 1).”.

Supplementary Fig. 1 Wavelength spectra of the visible light LEDs.

(5) Lines 93-94, Why the methanogenesis behaviour was different between anoxygenic photosynthetic bacteria and oxygenic photosynthetic bacteria under the light-dark conditions?

Response: We are sorry that the statements in the original manuscript were not clear. The differing methanogenesis behavior observed between anoxygenic and oxygenic photosynthetic bacteria under light-dark conditions might be attributed to the production of O₂ by oxygenic photosynthetic bacteria with H₂O as electron donors under light irradiation, thereby stimulating the

generation of ROS. These ROS, in turn, could oxidize organic matters to create potential carbon sources for biotic methanogenesis with methanogens in the dark. The relevant information has been replenished in the revised manuscript.

Revised in text: “This difference might be attributed to the production of O₂ by PCC6803 with H₂O as electron donors under illumination, thereby stimulating the generation of ROS. These ROS, in turn, could oxidize organic matters to create potential carbon sources for biotic methanogenesis with *M. b* in the dark (see detailed discussion below).”.

(6) Lines 97-98, Is it possible to provide more evidences to support the statement about the rapid growth and formation of a stable syntrophic coculture for methanogenesis?

Response: We thank the referee for pointing out this issue. Additional images depicting the rapid growth and formation of a stable syntrophic coculture were included in the Supporting Information part of the revised manuscript.

Revised in text: “The CH₄ production rate with PCC6803-*M. b*-Fe-EDTA continuously increased during the three successive cycles (Fig. 2c), contributing to the rapid growth and formation of a stable syntrophic coculture for methanogenesis (Supplementary Fig. 4).”.

Supplementary Fig. 4 Photographs depicting the evolving biofilm during the coculture. Scale bars: 1 μm .

(7) Lines 130-132, Can you show the change of biofilm during the coculture in a time scale with different pictures, maybe in Supporting Information? This may give the readers with more intuitive impression.

Response: Thank you for your suggestion. Additional images depicting the rapid growth and formation of biofilm during the coculture at different time points were included in the Supporting Information part of the revised manuscript.

Supplementary Fig. 4 Photographs depicting the evolving biofilm during the coculture. Scale bars: 1 μm .

(8) Lines 144-146, Maybe more discussion about the results of two-dimensional NMR should be conducted.

Response: As suggested, more discussion about the results of two-dimensional NMR has been replenished in the revised manuscript.

Revised in text: “These findings were further confirmed by two-dimensional NMR spectra, which provide the crucial through-bond correlations existing between coupled protons [two-dimensional gradient-selected homonuclear correlation spectroscopy (gCOSY) in Fig. 3e] and between protons and carbons via multiple-bond correlations [gradient-selected heteronuclear multiple bond correlation (gHMBC) in Fig. 3f] for each compound.²⁵ The ^1H and ^{13}C chemical shifts of lactate, pyruvate, methanol, and acetate correspond well to values from known library spectra (PubChem Database), such as the chemical shifts of ^1H (1.96 ppm) and ^{13}C (176.5 ppm) in acetate, confirming the presence of these compounds in aqueous solution during syntrophic methanogenesis.”

Reference

25. Griffith, E. C., Carpenter, B. K., Shoemaker, R. K., Vaida, V. Photochemistry of aqueous pyruvic acid. *P. Natl. Acad. Sci.* **110**, 11714-11719 (2013).

(9) Lines 165-170, Is it possible for *M. b* to directly use the electrons from the respiration process of PCC6803 in darkness for methanogenesis? I can understand that it may be very difficult to provide the direct evidences, but you can add the relevant discussion in the content.

Response: We are in agreement with the reviewer’s comment that *M. b* is possible to directly use the electrons from the respiration process of PCC6803 in darkness for methanogenesis. More detailed discussion has been replenished in the revised manuscript.

Revised in text: “In contrast, previous studies have shown that photosynthetic microorganisms can also generate an electrical current exclusively in darkness, using illumination as a recharge stage.^{35,36} To validate this, two-chamber H-cells were constructed, with PCC6803 and *M. b* separately inoculated into each chamber and then electrically connected by an external circuit, to mitigate the influence of direct electron exchange between *M. b* and PCC6803 on the photoelectron measurement. Similar results were observed that a continuous current was recorded during its dark fermentation period, albeit with lesser intensity compared to that in the light (**Supplementary Fig. 9b**). This finding indicates the potential DIET between PCC6803 and *M. b* via an electric syntrophic coculture in darkness.”

Supplementary Fig. 9 Current densities with PCC6803-*M. b*-Fe-EDTA and deletional controls. (a) in single-chamber microbial fuel cell, (b) in two-chamber H-cells. External potential bias: -0.5 V; working electrode: carbon cloth with a size of 3 cm × 3 cm; electrolyte: autotrophic medium; light on/off cycle: 4 h/20 h; light source: visible light LEDs. Data were recorded every 1 min by a data acquisition system (model 2700, Keithley Instruments, Ohio, USA).

References

35. Lin, C.-C., Wei, C.-H., Chen, C.-I., Shieh, C.-J., Liu, Y.-C. Characteristics of the photosynthesis microbial fuel cell with a *Spirulina platensis* biofilm. *Bioresour. Technol.* **135**, 640-643 (2012).
36. Fu, C.-C., Hung, T.-C., Wu, W.-T., Wen, T.-C., Su, C.-H. Current and voltage responses in instant photosynthetic microbial cells with *Spirulina platensis*. *Biochem. Eng. J.* **52**, 175-180 (2010).

(10) Lines 274-275, More information and discussion should be replenished for Fig. 5f.

Response: We thank the referee for pointing out this issue. As suggested, more detailed information and discussion for Fig. 5f has been replenished in the revised manuscript.

Revised in text: “On one hand, PCC6803 could synthesized organic compounds containing sulfur-bonded methyl groups, such as pyruvate and ethanol (Fig. 3g). Meanwhile, the existence of DMS was confirmed by gas chromatography-mass spectrometry (GC-MS) with an ion signal at $m/z = 62$, along with the retention time of 8.7 min in 400 MHz ^1H nuclear magnetic resonance (NMR) spectra (Fig. 5f). The DMS production might be partly attributed to the ROS oxidation of DMSP, because the DMSP was confirmed by the increasing intensity of DMS after alkali treatment for 12 h via the DMSP-to-DMS conversion.⁴² On the other hand, the possible release of the intracellular metabolites by *M. b.*, such as 2-(methylthio)ethanesulfonic acid ($\text{CH}_3\text{-S-CoM}$), might also serve as potential sources of methyl donors. These methyl donors could be oxidized by ROS for $\bullet\text{CH}_3$ formation, ultimately leading to abiotic CH_4 production.^{43,44}”.

References

42. Nagahata, T., Kajiwara, H., Ohira, S.-I., Toda, K. Simple field device for measurement of dimethyl sulfide and dimethylsulfoniopropionate in natural waters, based on vapor generation and chemiluminescence detection. *Anal. Chem.* **85**, 4461-4467 (2013).

43. Ernst, L., *et al.* Methane formation driven by reactive oxygen species across all living organisms. *Nature* **603**, 482-487 (2022).

44. Althoff, F., *et al.* Abiotic methanogenesis from organosulphur compounds under ambient conditions. *Nat. Commun.* **5**, 4205 (2014).

(11) Based on the discussion in the manuscript, it is better to adjust the order of different parameters in Figure 6 (First oxygenic photosynthetic bacteria, then anaerobic methanogenic archaea, and finally Fe-species). Meanwhile, the relevant experimental conditions should be more clearly clarified. For instance, which kind of Fe-species did you use when you investigated the influence of different oxygenic photosynthetic bacteria on the performance of syntrophic methanogenesis?

Response: We thank the referee for pointing out this issue. As suggested, the Figure 6 has been amended in the revised manuscript.

Fig. 6 | Common syntrophic methanogenesis by photosynthetic bacteria and methanogenic archaea.

(12) Line 335, The relevant information about the natural sunlight should be replenished.

Response: We are sorry that the statements in the original manuscript were not clear. As suggested, the relevant information about the natural sunlight has been replenished in the revised manuscript.

Revised in text: “The syntrophic methanogenesis experiments was also conducted on the roof of the Research Centre for Water Resources and Security Building at Fujian Agriculture and Forestry University in Fuzhou, China (latitude: 26.05 °N, longitude: 119.14 °E) under natural sunlight (from 08:00 to 20:00 with an average solar heat flux of $\sim 0.5 \text{ kW m}^{-2}$), with ambient temperatures ranging between 25 °C and 37 °C.”.

Reviewer #3 (Remarks to the Author):

This is a very interesting and very complex article. Stepping back and taking a broad perspective on the findings, the authors conclude from their experiments that a variety of methanogenic bacteria produce methane in the light and in the dark when co-cultured with a variety of different oxygenic phototrophs. More methane is made in the light (Fig. 2b), and Fe-EDTA considerably enhanced methane production. I suggest the authors choose among the good ideas they report here and build stronger experimental support for the ideas they want to prioritize. It would be a significant contribution to prove a general mechanism that supports a partnership between methanogens and oxygenic photoautotrophs. Presumably (but not explicitly clear in this manuscript), part of this partnership is the phototrophs benefit from the methanogens when they deplete oxygen by respiration. In addition to this concept, also in this manuscript are the concepts of direct interspecies electron transfer via Fe-EDTA, and abiotic methanogenesis via ROS. These three concepts all have some support from the reported experiments but none has compelling experimental support. In a manuscript with this many complex ideas and experiments, the supporting documentation can be very important for explaining concepts when page length limitations prevent digression. But, the supporting documents include interesting figures with very little explanation. Overall, I find the ideas in this manuscript interesting but the presentation of the data made it difficult to fully evaluate the experimental support the author's ideas. Although I think clarity and focus are a problem, I suspect with greater clarity and focus I would ask for more experiments. At this time I don't want to ask for more experiments when I'm not sure which if the findings are most central.

We are grateful for the valuable suggestions provided by the referee, which have greatly enhanced the quality of our manuscript. We have thoroughly revised the manuscript in accordance with these suggestions, and hope that our efforts have effectively addressed the concerns raised by the referee. The major modifications include:

(1) Additional data and discussion have been provided to demonstrated that oxygenic photoautotrophs were benefiting from the lowered hydrogen pressure in darkness; that is, the CO₂/H₂ methanogenesis of methanogens after oxygen depletion created more thermodynamically favourable conditions for the dark fermentation of oxygenic photoautotrophs.

(2) The Fe(III)/Fe(II) redox cycle with Fe-EDTA during syntrophic methanogenesis involves the reduction of Fe(III) by PCC6803, either through intracellular metabolism or extracellular electron transfer, and the oxidation of Fe(II) by ROS and O₂, rather than through DIET between PCC6803 and *M. b.*

(3) More detailed description and discussion of the supporting documents have been replenished in the revised manuscript.

1. The authors refer to the relationship between phototrophs and methanogen as a syntrophy but I did not notice evidence supporting the idea that the phototrophs benefited. In fact, in Fig 2e we see that the introduction of *Methanosarcina* (*M. b*) +Fe-EDTA seems to accelerate O₂ decline (note there is no PCC6803 control shown for comparison). My interpretation is that *M. b* is consuming oxygen, probably resulting in more rapid depletion of O₂ by the cyanobacteria when they are respiring in the dark, which likely will have negative consequences for the cyanobacteria. How does the cyanobacterium benefit? The impact appears to be negative. I recommend to not use the term "syntrophy" unless you can show benefits to the cyanobacterial partner. I appreciate that it is possible that the methanogens are making energy-yielding anaerobic metabolism more favorable for the cyanobacteria when O₂ is exhausted, by lowering H₂ partial pressures, but I found the discussion of that important mechanism too brief and unconvincing.

Response: We are sorry that the statements in the original manuscript were not clear. *M. b* is a strictly anaerobic methanogen that cannot consume oxygen (*Nat. Rev. Microbiol.* 7, 568-577 (2019)). The anaerobic microenvironment in darkness during syntrophic methanogenesis was created by the synergistic interaction of the respiratory process of PCC6803 and the redox cycle of EDTA-Fe (Fig. 2e). Under such anaerobic conditions, as pointed out by the reviewer, the methanogenesis metabolism of *M. b* indeed created more thermodynamically favorable conditions for PCC6803. This is because PCC6803 contains a single [NiFe]-hydrogenase, HoxEFUYH, which is involved in fermentative hydrogen production as well as working as an electron valve when photosynthesis resumes under anaerobic conditions (*Int. J. Hydrogen Energ.* 27, 1229-1237 (2002)). Notably, HoxEFUYH works bidirectionally with a bias to proton reduction rather than hydrogen oxidation (*J. Am. Chem. Soc.* 133, 11308-11319 (2011)). Therefore, an increase in hydrogen pressure due to hydrogen accumulation during dark fermentation might result in a significant decrease in hydrogenase activity, thereby influencing the growth and metabolism of PCC6803 (*Sci. Rep.* 5, 12424 (2015)). This inference was confirmed by the lower chlorophyll concentration, quantum yield of PSII primary photochemical reactions (Fv/Fm), and copy number of *cpcG* in bare PCC6803, along with a higher H₂ concentration compared with syntrophic methanogenesis (Supplementary Fig. 8). The superior activity of PCC6803 during syntrophic methanogenesis were attributed to the versatile metabolic pathways of *M. b*, which significantly lowered hydrogen pressure via CO₂/H₂ methanogenesis (Supplementary Fig. 7), and created more thermodynamically favourable conditions for the dark fermentation of PCC6803. The relevant information has been replenished in the revised manuscript.

Revised in text: "However, it should be pointed out that PCC6803 contains a single [NiFe]-hydrogenase, HoxEFUYH, which is involved in fermentative hydrogen production as well as working as an electron valve when photosynthesis resumes under anaerobic conditions.²⁹ Notably, HoxEFUYH works bidirectionally with a bias to proton reduction rather than hydrogen oxidation.³⁰ Therefore, an

increase in hydrogen pressure due to hydrogen accumulation during dark fermentation might result in a significant decrease in hydrogenase activity, thereby influencing the growth and metabolism of PCC6803.³¹ This inference was confirmed by the lower chlorophyll concentration, quantum yield of PSII primary photochemical reactions (Fv/Fm), and copy number of *cpcG* in bare PCC6803, along with a higher H₂ concentration compared with syntrophic methanogenesis (Supplementary Fig. 8). The superior activity of PCC6803 during syntrophic methanogenesis were attributed to the versatile metabolic pathways of *M. b.*, which significantly lowered hydrogen pressure via CO₂/H₂ methanogenesis (Supplementary Fig. 7), and created more thermodynamically favourable conditions for the dark fermentation of PCC6803.³²

Supplementary Fig. 8 (a) Chlorophyll concentration, (b) quantum yield of PSII primary photochemical reactions (Fv/Fm), and (c) copy number of *cpcG* in bare PCC6803 and PCC6803-*M. b.*-Fe-EDTA under a light-dark cycle of 4 h-20 h. Different letters represent statistically significant difference ($p < 0.05$) in different groups.

References

29. Laurent, C., Florence, M., Laetitia, B., Geneviève, G., Paulette, V., Gilles, P. Limiting steps of hydrogen production in *Chlamydomonas reinhardtii* and *Synechocystis* PCC 6803 as analysed by light-induced gas exchange transients. *Int. J. Hydrogen Energ.* **27**, 1229-1237 (2002).
30. McIntosh, C. L., Germer, F., Schulz, R., Appe, J., Jones, A. K. The [NiFe]-Hydrogenase of the Cyanobacterium *Synechocystis* sp. PCC 6803 works bidirectionally with a bias to H₂ production. *J. Am. Chem. Soc.* **133**, 11308-11319 (2011).
31. De Rosa, E., *et al.* [NiFe]-hydrogenase is essential for cyanobacterium *Synechocystis* sp. PCC 6803 aerobic growth in the dark. *Sci. Rep.* **5**, 12424 (2015).

Stams, A. J., Plugge, C. M. Electron transfer in syntrophic communities of anaerobic bacteria and

archaea. *Nat. Rev. Microbiol.* **7**, 568-577 (2019).

2. A rather straightforward, conventional explanation for methanogenic activity would be that the methanogens, in the absence of other cell types except the photoauxotrophs, are benefiting from photosynthetic organic matter production, performing methanogenesis when oxygen is depleted by cyanobacterial respiration in the dark. While not surprising from a theoretical perspective, this potentially is an interesting finding. What percentage of gross primary production was diverted to methane formation? Are your findings a manifestation of common microbiological activities, or is there a co-evolved, specific interaction occurring? I found myself unable to decide which was the case for the data provided. I recommend that you estimate the percentage of gross primary production accounted for by methane production. Is there evidence that anaerobic PCC6308 can use fermentative metabolism to produce energy in the absence of oxygen, which would indicate they might benefit from the methanogens creating thermodynamically favorable conditions for fermentation?

Response: We appreciate the reviewer's valuable suggestion. Indeed, there exists a co-evolved, specific interaction during syntrophic methanogenesis by oxygenic photosynthetic bacteria and anaerobic methanogenic archaea in darkness. Specifically, *M. b*, in the absence of other cell types except PCC6803, were benefiting from photosynthetic organic matter production. It was estimated that 5% of gross primary production was diverted to CH₄ formation. Meanwhile, due to the CO₂/H₂ methanogenesis of *M. b*, PCC6803 were benefiting from the lowered hydrogen pressure, creating more thermodynamically favourable conditions for the dark fermentation of PCC6803. The relevant information has been replenished in the revised manuscript.

Revised in text: “In conclusion, besides the abiotic methanogenesis under illumination, there exists a co-evolved, specific interaction during syntrophic methanogenesis by oxygenic photosynthetic bacteria and anaerobic methanogenic archaea in darkness. Specifically, *M. b*, in the absence of other cell types except PCC6803, were benefiting from photosynthetic organic matter production. It was estimated that 40% of gross primary production was diverted to CH₄ formation. Meanwhile, due to the CO₂/H₂ methanogenesis of *M. b*, PCC6803 were benefiting from the lowered hydrogen pressure, creating more thermodynamically favourable conditions for the dark fermentation of PCC6803.”.

3. The manuscript indicates that Fe-EDTA supports methanogenesis, presumably by serving as a carrier for electrons DIET. Can you prove that PCC6803 reduces Fe-EDTA and *M. b* oxidizes it?

Response: We are sorry that the description in Figure 1 is unclear that has led to misunderstandings. Our additional experiments indicated that *M. b* could not oxidize Fe-EDTA (data no shown). The Fe(III)/Fe(II) redox cycle during the syntrophic methanogenesis involves the Fe(III)

reduction by PCC6803 (Supplementary Fig. 13), either through intracellular metabolism (*Bioelectrochemistry* 105, 103-109 (2015)) or extracellular electron transfer (as indicated by photocurrent production), and Fe(II) oxidation by ROS and O₂, rather than through DIET between PCC6803 and *M. b*. We have revised the Figure 1 and included additional experimental data regarding the reduction of Fe-EDTA by PCC6803 in the revised manuscript.

Revised in text: “The produced Fe(III) through the oxidation of ROS and O₂ could be reduced by PCC6803 (Supplementary Fig. 13), either through intracellular metabolism or extracellular electron transfer, and finally established an effective Fe(III)/Fe(II) redox cycle.”.

Reference

Thorne, R. J., Schneider, K., Hu, H., Cameron, P. J. Iron reduction by the cyanobacterium *Synechocystis* sp. PCC 6803. *Bioelectrochemistry* **105**, 103-109 (2015).

Fig. 1 | Schematic illustration of methanogenesis by synergistic interaction between oxygenic photosynthetic bacteria and anaerobic methanogenic archaea.

Supplementary Fig. 13 The reduction of Fe-EDTA by PCC6803.

4. Figure 4. “Mechanisms of biotic methanogenesis with PCC6803-M. *b*-Fe-EDTA as revealed by transcriptomic analyses”. It’s confusing to show ROS in this figure but no role for Fe-EDTA, which is purportedly important for biological methane production.

Response: We thank the referee for pointing out this issue. As suggested, Fe-EDTA and the transcript levels of related genes encoding the Fe-EDTA conversion have been replenished in the revised manuscript.

Revised in text: “Remarkably, the iron uptake and transport (ferric uptake (Fut) and ferrous iron transport (Feo) systems) were also enhanced with higher gene expression, particularly in the light period, facilitating the Fe redox process during syntrophic methanogenesis.”.

Fig. 4 | Mechanisms of biotic methanogenesis with PCC6803-*M. b*-Fe-EDTA as revealed by transcriptomic analyses.

5. In Fig.2, oxygen production appears to continue for 5 days (2e) but methane production falls off sharply after day 2. Is this the correct interpretation of the data? Why does methane production stop?

Response: Yes, the interpretation of the data is correct. The CH₄ production rate with PCC6803-*M. b*-Fe-EDTA slowed down somewhat after day 2 (**Fig. 2b**), despite the oxygen production appears to continue. This result might be attributed to the fact that the growth of PCC6803 led to an enhanced photosynthetic oxygen evolution, elevating DO accumulation in the light and extending the time required to establish an anaerobic microenvironment for biotic methanogenesis in darkness (**Fig. 2e**). As a result, the rate of CH₄ production with PCC6803-*M. b*-Fe-EDTA slowed down somewhat after day 2 but persisted (**Fig. 2b**). The relevant information has been replenished in the revised manuscript.

Revised in text: “However, the growth of PCC6803 led to an enhanced photosynthetic oxygen evolution, elevating DO accumulation in the light and extending the time required to establish an anaerobic microenvironment for biotic methanogenesis in darkness (**Fig. 2e**). As a result, the rate of CH₄ production with PCC6803-*M. b*-Fe-EDTA slowed down somewhat after day 2 but persisted (**Fig.**

2b)”.

6. There is almost no explanation of the term 'photocurrent', and the description of these experiments is too brief. If the photocurrent is through a graphite substrate, does this mean its competing with electron transfer to *M. b*?

Response: We are sorry that the statements about photocurrent were not detailed enough in the original manuscript. Specifically, the photocurrent measurement was initially conducted in a single-chamber microbial fuel cell, using the carbon cloth with a size of 3 cm × 3 cm as the working electrodes. A distinct photocurrent was observed under illumination (Supplementary Fig. 9a), indicating that PCC6803 can release electrons extracellularly. Although *M. b* served as electron acceptors capable of accepting photoelectrons from PCC6803 for CO₂ reduction (*Nat. Commun.* 13, 6612 (2022); *Angew. Chem. Int. Ed.* 61, e202206508 (2022)), the syntrophic methanogenesis via a direct interspecies electron transfer (DIET) pathway was less likely to occur with PCC6803-*M. b*-Fe-EDTA under illumination. This was because that the growth and metabolism of *M. b* is highly sensitive to the oxygen exposure during the photosynthetic oxygen evolution. In contrast, previous studies have shown that photosynthetic microorganisms can also generated an electrical current exclusively in darkness, using illumination as a recharge stage (*Bioresour. Technol.* 135, 640-643 (2012); *Biochem. Eng. J.* 52, 175-180 (2010)). To validate this, two-chamber H-cells were constructed. PCC6803 and *M. b* separately inoculated into each chamber and then electrically connected by an external circuit, to mitigate the influence of direct electron exchange between *M. b* and PCC6803 on the photoelectron measurement. Similar results were observed that a continuous current was recorded during its dark respiratory period, albeit with lesser intensity compared to that in the light (Supplementary Fig. 9b). This finding indicates the potential DIET between PCC6803 and *M. b* via an electric syntrophic coculture in darkness. We have now provided a detailed method for photocurrent measurement and relevant discussion in the revised manuscript.

Revised in text: “A distinct photocurrent was also observed under illumination during syntrophic methanogenesis (Supplementary Fig. 9a), indicating that PCC6803 can release electrons extracellularly. Although *M. b* served as electron acceptors capable of accepting photoelectrons from PCC6803 for CO₂ reduction ($\text{CO}_2 + 8\text{e}^- + 8\text{H}^+ \rightarrow \text{CH}_4 + 2\text{H}_2\text{O}$),^{32,33} the syntrophic methanogenesis via a direct interspecies electron transfer (DIET) pathway was less likely to occur with PCC6803-*M. b*-Fe-EDTA under illumination. This was because that the growth and metabolism of *M. b* is highly sensitive to the oxygen exposure during the photosynthetic oxygen evolution.”.

“In contrast, previous studies have shown that photosynthetic microorganisms can also generated an electrical current exclusively in darkness, using illumination as a recharge stage.^{35,36} To validate

this, two-chamber H-cells were constructed, with PCC6803 and *M. b* separately inoculated into each chamber and then electrically connected by an external circuit, to mitigate the influence of direct electron exchange between *M. b* and PCC6803 on the photoelectron measurement. Similar results were observed that a continuous current was recorded during its dark fermentation period, albeit with lesser intensity compared to that in the light (**Supplementary Fig. 9b**). This finding indicates the potential DIET between PCC6803 and *M. b* via an electric syntrophic coculture in darkness.”.

“The photocurrents (*I-t*) were initially measured in a single-chamber microbial fuel cell (MFC) by applying an external potential bias of -0.5 V. Carbon cloth with a size of 3 cm × 3 cm was used as the working electrodes, and autotrophic medium with EDTA-Fe was employed as electrolyte. The cell had a liquid volume of 200 mL and a headspace volume of 150 mL. Data were recorded every 1 min by a data acquisition system (model 2700, Keithley Instruments, Ohio, USA). In addition, photocurrent measurement was also conducted under identical condition, but using two-chamber H-cells instead of the single-chamber MFC.¹⁰ A proton exchange membrane (Nafion 117, DuPont Co., USA) was used to separate the anodic and cathodic chambers, along with autotrophic medium as electrolyte.”.

Supplementary Fig. 9 Current densities with PCC6803-*M. b*-Fe-EDTA and deletional controls. (a) in single-chamber microbial fuel cell, (b) in two-chamber H-cells. External potential bias: -0.5 V; working electrode: carbon cloth with a size of 3 cm × 3 cm; electrolyte: autotrophic medium; light on/off cycle: 4 h/20 h; light source: visible light LEDs. Data were recorded every 1 min by the MultiPalmSens4 Potentiostat (PalmSens, The Netherlands).

References

32. Ye, J., *et al.* Solar-driven methanogenesis with ultrahigh selectivity by turning down H₂

production at biotic-abiotic interface. *Nat. Commun.* **13**, 6612 (2022).

33. Hu, A., Ye, J., Ren, G., Qi, Y., Chen, Y., Zhou, S. Metal-free semiconductor-based bio-nano hybrids for sustainable CO₂-to-CH₄ conversion with high quantum yield. *Angew. Chem. Int. Ed.* **61**, e202206508 (2022).

35. Lin, C.-C., Wei, C.-H., Chen, C.-I., Shieh, C.-J., Liu, Y.-C. Characteristics of the photosynthesis microbial fuel cell with a *Spirulina platensis* biofilm. *Bioresour. Technol.* **135**, 640-643 (2012).

36. Fu, C.-C., Hung, T.-C., Wu, W.-T., Wen, T.-C., Su, C.-H. Current and voltage responses in instant photosynthetic microbial cells with *Spirulina platensis*. *Biochem. Eng. J.* **52**, 175-180 (2010).

7. Legend, Fig. 2. Indicate that grey means dark, if that is correct.

Response: Yes, the grey color in Fig. 2 presents the dark period during the syntrophic methanogenesis under a light-dark cycle of 4 h-20 h. The relevant information has been added to the legend of Fig. 2 in the revised manuscript.

Revised in text: “The grey color in **b** and **e** presents the dark period during the syntrophic methanogenesis under a light-dark cycle of 4 h-20 h. Error bars represent the standard deviation of $n = 3$ analytical replicates from one experiment. Different letters represent statistically significant difference ($p < 0.05$) in different groups.”.

8. The legends for the supporting figures are too brief. Please add details.

Response: We thank the referee for pointing out this issue. More information has been added to the legends of the supporting figures in the revised manuscript.

Revised in text: “**Supplementary Fig. 1** Wavelength spectra of the visible light LEDs.

Supplementary Fig. 3 CH₄ yield with PCC6803-*M. b*-Fe-EDTA and deletional controls. “+/-” symbols represent the presence or absence of the relevant component. Error bars represent the standard deviation of $n = 3$ analytical replicates from one experiment. Different letters represent statistically significant difference ($p < 0.05$) in different groups.

Supplementary Fig. 4 Photographs depicting the evolving biofilm during the coculture. Scale bars: 1 μm .

Supplementary Fig. 5 Variation of biofilm thickness with PCC6803-*M. b*-Fe-EDTA after 18 days of syntrophic coculturing. The green fluorescence represents live *M. b*, while the red fluorescence represents PCC6803 and dead *M. b*, respectively.

Supplementary Fig. 6 H₂ yield with PCC6803-*M. b*-Fe-EDTA and deletional controls. Error bars represent the standard deviation of n = 3 analytical replicates from one experiment.

Supplementary Fig. 7 Variation of H₂ and CH₄ yields with PCC6803-*M. b*-Fe-EDTA after SBES addition. Error bars represent the standard deviation of n = 3 analytical replicates from one experiment. SBES, sodium 2-bromoethanesulfonate.

Supplementary Fig. 8 (a) Chlorophyll concentration, (b) quantum yield of PSII primary photochemical reactions (Fv/Fm), and (c) copy number of *cpcG* in bare PCC6803 and PCC6803-*M. b*-Fe-EDTA under a light-dark cycle of 4 h-20 h. Different letters represent statistically significant difference ($p < 0.05$) in different groups.

Supplementary Fig. 9 Current densities with PCC6803-*M. b*-Fe-EDTA and deletional controls. (a) in single-chamber microbial fuel cell, (b) in two-chamber H-cells. External potential bias: -0.5 V; working electrode: carbon cloth with a size of 3 cm × 3 cm; electrolyte: autotrophic medium with/without EDTA-Fe; light on/off cycle: 4 h/20 h; light source: visible light LEDs. Data were recorded every 1 min by the MultiPalmSens4 Potentiostat (PalmSens, The Netherlands).

Supplementary Fig. 10 Transcriptomic analyses of key genes encoding Calvin cycle. *glk*, glucokinase; *pgi*, glucose-6-phosphate isomerase; *fbp*, fructose 1,6-bisphosphatase; *glp*, GlpX protein; *fda*, fructose-bisphosphate aldolase; *gap*, glyceraldehyde-3-phosphate dehydrogenase; *pgk*, phosphoglycerate kinase; *zwf*, glucose 6-phosphate dehydrogenase; *dev*, glucose-6-P-dehydrogenase; *tkt*, transketolase; *rpi*, ribose 5-phosphate isomerase; *tal*, transaldolase; *prk*, phosphoribulokinase; *rbc*, ribulose-1,5-bisphosphate carboxylase; *gnd*, 6-phosphogluconate dehydrogenase.

Supplementary Fig. 11 Transcriptomic analyses of key genes encoding TCA cycle. *glt*, citrate synthase; *mdh*, malate dehydrogenase; *fum*, fumarase; *sdh*, succinate dehydrogenase iron-sulfur protein; *suc*, succinyl-CoA synthetase; *pdh*, dihydrolipoamide dehydrogenase; *icd*, isocitrate dehydrogenase; *acn*, aconitate hydratase.

Supplementary Fig. 13 The reduction of Fe-EDTA by PCC6803.

Supplementary Fig. 14 Effects of different ROS quenching reagents on methanogenesis performance after 18 days of syntrophic coculturing. Error bars represent the standard deviation of n = 3 analytical replicates from one experiment. Different letters represent statistically significant difference ($p < 0.05$) in different groups. TBA, tert-butyl alcohol; SOD, superoxide dismutase.

Supplementary Fig. 15 Transcriptomic analyses of key antioxidant stress genes in *M. b* and PCC6803 during the syntrophic methanogenesis. Sod, superoxide dismutase; Kat, catalase; sll1621, type II peroxiredoxins; AhpC and AhpE, alkylhydroperoxidases; Htp and Hsp, heat shock proteins; Rbr,

rubrerythrin; Dps, DNA-binding protein from starved cells.

Supplementary Fig. 16 Periodic variation of dissolved oxygen concentration under varied light intensities and illumination times (illumination time of 8 hours with the light intensity of 7 W/m², and illumination time of 12 hours with the light intensity of 2 W/m²). The grey color presents the dark period during the syntrophic methanogenesis.

Supplementary Fig. 17 CH₄ yield with PCC6803-*M. b*-Fe-EDTA by natural sunlight irradiation from 08:00 to 20:00 with an average solar heat flux of ~0.5 kW m⁻² and ambient temperatures ranging between 25 °C and 37 °C. Error bars represent the standard deviation of n = 3 analytical replicates from one experiment.”.

REVIEWER COMMENTS

Reviewer #1 (Remarks to the Author):

The authors have thoroughly addressed the comments of the three reviewers by providing comprehensive responses and making the necessary modifications. As a result, the quality of the current paper has substantially improved. Hence, I propose accepting it.

Reviewer #2 (Remarks to the Author):

I am satisfied with the revised manuscript. The authors carefully addressed all comments.

Reviewer #3 (Remarks to the Author):

This manuscript reports a diverse collection of experiments that collectively support the conclusion that methanogens and cyanobacteria may interact synergistically, benefitting both cell types in dense co-cultures on light dark cycles by providing methanogens with a source of electrons for methanogenic energy production and cyanobacteria with an electron sink that increases the energy yield of fermentative metabolism of stored carbohydrates during dark periods. Theoretically, such an association makes sense, and, because of the prevalence of cyanobacterial mats, it could potentially have a broad impact by widespread methane production. However, most of the methane production by the culture occurs in the light (Fig. 2b), which points to the proposed abiotic mechanisms of methane production. But, that does not entirely make sense because it's not clear why methanogenic bacteria would be required for that process. Does PCC6803 make methane in the light without the other factors? This is an important control that I did not find. If not, does this not diminish the argument that production of methane in the light is due to abiotic mechanisms?

The syntrophy idea is better supported in this version of the manuscript, but the role of methanogenic bacteria in abiotic methane production needs further support. The authors modified their manuscript to address one of my foremost criticisms, which was the lack of clarity in the original manuscript about the mechanisms by which methanogenesis benefits the cyanobacteria.

The mechanisms reported are multiple and complex, which makes the narrative difficult to follow. Although this is a challenge throughout the manuscript, figure 1, a summary figure, does not help and should be altered (and perhaps the legend needs to change), for clarity. The figure shows oxygenic photosynthetic bacteria in the light and anaerobic methanogenic bacteria in the dark, but they are both together all the time in the light and the dark, so I think this is not a good way to illustrate the concepts. Rather, I suggest that you show them both together in the dark and the light, and use the figure to illustrate how the proposed mechanisms of methane production and interactions of these two cell types change between light and dark.

The manuscript give the impression that across aquatic environments these processes are occurring. The proposed syntrophy only occurs in micro environments that periodically become anoxic. At some positions in the manuscript this is clear, but in the text explaining the methane paradox this point needs clarification.

My overall impression is that this manuscript is not yet ready to publish. Very complex ideas are advanced to explain the observations. This makes the manuscript a challenge to write, because a variety of different processes that are proposed must be proven. It's also a challenge to read, because of the multiple interpretations for methane production.

Re: NCOMMS-24-04041A

Point-by-point responses to Referees' Comments

Reviewer #1 (Remarks to the Author):

The authors have thoroughly addressed the comments of the three reviewers by providing comprehensive responses and making the necessary modifications. As a result, the quality of the current paper has substantially improved. Hence, I propose accepting it.

Response: We thank the reviewer very much for the kind comments.

Reviewer #2 (Remarks to the Author):

I am satisfied with the revised manuscript. The authors carefully addressed all comments.

Response: We thank the reviewer very much for the kind comments.

Reviewer #3 (Remarks to the Author):

1) This manuscript reports a diverse collection of experiments that collectively support the conclusion that methanogens and cyanobacteria may interact synergistically, benefitting both cell types in dense co-cultures on light-dark cycles by providing methanogens with a source of electrons for methanogenic energy production and cyanobacteria with an electron sink that increases the energy yield of fermentative metabolism of stored carbohydrates during dark periods. Theoretically, such an association makes sense, and, because of the prevalence of cyanobacterial mats, it could potentially have a broad impact by widespread methane production. However, most of the methane production by the culture occurs in the light (Fig. 2b), which points to the proposed abiotic mechanisms of methane production. But, that does not entirely make sense because it's not clear why methanogenic bacteria would be required for that process. Does PCC6803 make methane in the light without the other factors? This is an important control that I did not find. If not, does this not diminish the argument that production of methane in the light is due to abiotic mechanisms?

Response: We thank the reviewer for pointing out this issue. The relevant results indicated that the O₂ produced by PCC6803 during the photosynthetic oxygen evolution significantly stimulated the ROS production by methanogenic bacteria (*M. b*) in illumination, thereby enhancing the abiotic CH₄ production process (as detailed in the next comment). Moreover, additional experiments showed that PCC6803 produced almost no CH₄ under illumination without the presence of other factors (**Fig. 2b**), suggesting the potential existence of an abiotic CH₄ production process. The relevant information has been replenished in the revised manuscript.

Revised in text: “Substantial CH₄ production during the light period was observed with PCC6803-*M. b*-Fe-EDTA (**Fig. 2b**). However, deletional control experiment revealed that PCC6803 produced almost no CH₄ under illumination without the presence of other factors (**Fig. 2b**), suggesting the potential existence of an abiotic CH₄ production process.”.

Fig. 2 | Light-driven methanogenesis in a light-dark cycle. b, Typical time course of CH₄ yield in the first 4 days.

2) The syntrophy idea is better supported in this version of the manuscript, but the role of methanogenic bacteria in abiotic methane production needs further support. The authors modified their manuscript to address one of my foremost criticisms, which was the lack of clarity in the original manuscript about the mechanisms by which methanogenesis benefits the cyanobacteria.

Response: The relevant experiments indicated that the O₂ produced by PCC6803 during the photosynthetic oxygen evolution significantly stimulated the ROS production by methanogenic bacteria (*M. b*) in illumination, thereby enhancing the abiotic CH₄ production process. This inference was confirmed through the ROS production experiments with *M. b* under varying O₂ concentrations. It was found that higher O₂ concentrations led to the production of more ROS, such as •OH and H₂O₂ (**Supplementary Fig. 13**). Stable isotope analysis with ¹⁸O₂ further confirmed that the produced ROS stemmed from O₂ reduction, evidenced by the observed 5,5-dimethyl-1-pyrroline-N-oxide (DMPO)-¹⁸OH ($m/z = 132.09$, **Supplementary Fig. 14**). Previous studies have shown that when anaerobic cells were exposed to oxygen-rich environments, molecular O₂ adventitiously abstracted electrons from the reduced flavins or metal centers of some redox enzymes, resulting in the ROS formation (*mBio* 8(1), e01873-16 (2017); *Mol. Microbiol.* 75(6), 1389-1401 (2010); *J. Biol. Chem.* 266(11), 6957-6965 (1991)). As these events relied on collision frequency, the ROS production rate was directly proportional to the O₂ concentration (*Nat. Rev. Microbiol.* 19(12), 774-785 (2021)). Subsequently, the produced ROS were inadvertently released extracellularly (*Biochem. Biophys. Res. Commun.* 36(6), 891-897

(1969)). Along with the potentially produced ROS by Fenton reaction with Fe-EDTA and/or PCC6803 during the photochemical energy conversion for bioenergetic production (*P. Natl. Acad. Sci.* 102, 6502-6507 (2005)), various methyl donors were oxidized by ROS to form methyl radicals ($\bullet\text{CH}_3$) as intermediates that eventually resulted in abiotic methanogenesis under oxic conditions. The relevant information has been replenished in the revised manuscript.

Revised in text: “The concentrations of these ROS, particularly $\bullet\text{OH}$ and H_2O_2 , significantly increased under illumination (**Fig. 5d**), likely attributed to the produced O_2 by PCC6803 during the photosynthetic oxygen evolution that significantly stimulated the ROS production by *M. b*. This inference was confirmed through the ROS production experiments with *M. b* under varying O_2 concentrations. It was found that higher O_2 concentrations led to the production of more ROS, such as $\bullet\text{OH}$ and H_2O_2 (**Supplementary Fig. 13**). Stable isotope analysis with $^{18}\text{O}_2$ further confirmed that the produced ROS stemmed from O_2 reduction, evidenced by the observed 5,5-dimethyl-1-pyrroline-N-oxide (DMPO)- ^{18}OH ($m/z = 132.09$, **Supplementary Fig. 14**). Previous studies have shown that when anaerobic cells were exposed to oxygen-rich environments, molecular O_2 adventitiously abstracted electrons from the reduced flavins or metal centers of some redox enzymes, resulting in the ROS formation ($\text{O}_2 \rightarrow \bullet\text{O}_2^- \rightarrow \text{H}_2\text{O}_2 \rightarrow \bullet\text{OH}$).⁴⁰⁻⁴² As these events relied on collision frequency, the ROS production rate was directly proportional to the O_2 concentration.⁴³ Subsequently, the produced ROS were inadvertently released extracellularly.⁴⁴”.

“ROS production experiments with *M. b* under varying O_2 concentrations (0.0%, 0.5% and 1.0%) were conducted to evaluate the role of *M. b* in abiotic CH_4 production under illumination. Meanwhile, stable isotope experiments using ^{18}O -labeled O_2 were carried out to further determine the origin of the produced $\bullet\text{OH}$. The DMPO- ^{18}OH , formed through the interaction between DMPO and $\bullet^{18}\text{OH}$, was identified using an ultrahigh-performance liquid chromatography-triple quadrupole mass spectrometer (LC-MS, TSQ Endura, Thermo Fisher, USA).⁵⁹”.

Supplementary Fig. 13 Effects of O₂ concentration on the production of •OH and H₂O₂ by *M. b.* Error bars represent the standard deviation of n = 3 analytical replicates from one experiment. Different letters represent statistically significant difference ($p < 0.05$) in different groups.

Supplementary Fig. 14 Mass spectra of DMPO-•OH with ¹⁸O₂ as reaction species during ROS production by *M. b.* The mass-to-charge ratios (m/z) of 130.09 and 132.09 represent DMPO-¹⁶OH and DMPO-¹⁸OH, respectively.

References

40. Lu, Z., Imlay, J. A. The fumarate reductase of *Bacteroides thetaiotaomicron*, unlike that of *Escherichia coli*, is configured so that it does not generate reactive oxygen species. *mBio* **8**(1), e01873-16 (2017).
41. Korshunov, S., Imlay, J. A. (2010). Two sources of endogenous hydrogen peroxide in *Escherichia coli*. *Mol. Microbiol.* **75**(6), 1389-1401 (2010).
42. Imlay, J. A., Fridovich, I. Assay of metabolic superoxide production in *Escherichia coli*. *J. Biol. Chem.* **266**(11), 6957-6965 (1991).
43. Lu, Z., Imlay, J. A. When anaerobes encounter oxygen: mechanisms of oxygen toxicity, tolerance and defence. *Nat. Rev. Microbiol.* **19**(12), 774-785 (2021).
44. Massey, V., *et al.* The production of superoxide anion radicals in the reaction of reduced flavins and flavoproteins with molecular oxygen. *Biochem. Biophys. Res. Co.* **36**(6), 891-897 (1969).
59. Huang, L., Liu, X., Rensing, C., Yuan, Y., Zhou, S., Nealon, K. H. Light-independent anaerobic microbial oxidation of manganese driven by an electrosynthetic coculture. *ISME J.* **17**, 163-171 (2022).

3) *The mechanisms reported are multiple and complex, which makes the narrative difficult to follow. Although this is a challenge throughout the manuscript, figure 1, a summary figure, does not help and should be altered (and perhaps the legend needs to change), for clarity. The figure shows oxygenic photosynthetic bacteria in the light and anaerobic methanogenic bacteria in the dark, but they are both together all the time in the light and the dark, so I think this is not a good way to illustrate the concepts. Rather, I suggest that you show them both together in the dark and the light, and use the figure to illustrate how the proposed mechanisms of methane production and interactions of these two cell types change between light and dark.*

Response: We appreciate the reviewer's valuable suggestion. As suggested, the Figure 1 and its legend have been revised, and the relevant information has been replenished in the revised manuscript.

Revised in text: “The results showed that CH₄ production by the interaction of oxygenic photosynthetic bacteria and anaerobic methanogenic archaea was significantly enhanced through the redox cycling of Fe-ethylenediaminetetraacetic acid (Fe-EDTA), involving both syntrophic methanogenesis and abiotic methanogenesis during the periodic dark-light cycles (Fig. 1). Specifically, in darkness, the organics and H₂ produced by PCC6803 during dark fermentation were utilized as carbon sources and reducing equivalents by *M. b* for syntrophic methanogenesis under anoxic conditions. The significantly lowered hydrogen pressure by *M. b*, in turn, created more

thermodynamically favourable conditions for PCC6803. In contrast, in illumination, the photosynthesized organic compounds and their intermediates by PCC6803 served as potential methyl donors ($-\text{CH}_3$). Meanwhile, the simultaneously produced O_2 stimulated reactive oxygen species (ROS) production by *M. b.* Along with the Fenton reaction with Fe-EDTA, various methyl donors were oxidized by ROS to form methyl radicals ($\bullet\text{CH}_3$) as intermediates that eventually resulted in abiotic methanogenesis under oxic conditions”.

Fig. 1 | Schematic illustration of CH_4 production by the interaction of oxygenic photosynthetic bacteria and anaerobic methanogenic archaea through both syntrophic methanogenesis (under anoxic conditions in darkness) and abiotic methanogenesis (under oxic conditions in illumination) pathways during the periodic light-dark cycles.

4) *The manuscript gave the impression that across aquatic environments these processes are occurring. The proposed syntrophy only occurs in micro environments that periodically become anoxic. At some positions in the manuscript this is clear, but in the text explaining the methane paradox this point needs clarification.*

Response: We agree that the statement in the original manuscript is not clear. As depicted in the revised Figure 1 of the manuscript, CH₄ production by oxygenic photosynthetic bacteria and methanogenic archaea involved both syntrophic methanogenesis (under anoxic conditions in darkness) and abiotic methanogenesis (under oxic conditions in illumination) pathways during the periodic light-dark cycles, which potentially explained the prevalent methane paradox phenomenon. The relevant information has been replenished in the revised manuscript.

Revised in text: “Thus, this light-driven methanogenesis process, involved both syntrophic methanogenesis (under anoxic conditions in darkness) and abiotic methanogenesis (under oxic conditions in illumination) during the periodic dark-light cycles, surpasses the conventional methane production pathways (i.e., acetoclastic methanogenesis and hydrogenotrophic methanogenesis), and potentially making a more significant contribution to the global CH₄ cycle. The inference was supported by the correlation between CH₄ supersaturation and photosynthesis.^{8,9,54}”

5) *My overall impression is that this manuscript is not yet ready to publish. Very complex ideas are advanced to explain the observations. This makes the manuscript a challenge to write, because a variety of different processes that are proposed must be proven. It's also a challenge to read, because of the multiple interpretations for methane production.*

Response: We agree with the reviewer that the relevant mechanisms explaining the observations are somewhat complex. Like many other studies for understanding (different) complex processes, this study would serve as an initial step to reveal this complex process in the nature. To better illustrate the methanogenesis process by oxygenic photosynthetic bacteria and anaerobic methanogenic archaea, we have restructured the explanation into two main pathways during the periodic light-dark cycles, based on the obtained results and reviewer’s suggestions, involving both syntrophic methanogenesis (under anoxic conditions in darkness) and abiotic methanogenesis (under oxic conditions in illumination). Accordingly, we have modified the article’s title, structure and figures, including the revision of the relevant figures (e.g., Fig. 1 and Fig. 5a) and expansion of the related discussion (e.g., the role of *M. b* during abiotic methanogenesis). We sincerely hope that our revisions have satisfactorily addressed the referee’s concerns.